# Molecular and anatomical characterization of parabrachial neurons and their axonal projections

**Jordan L Pauli[1†], Jane Y Chen[1*†], Marcus L Basiri[2,3†], Sekun Park[1], Matthew E Carter[1], Elisenda Sanz[3], G Stanley McKnight[3], Garret D Stuber[2,3], Richard D Palmiter[1*]**

[1]Department of Biochemistry, Howard Hughes Medical Institute, University of Washington, Seattle, United States; [2]Center for the Neurobiology of Addiction, Pain, and Emotion, Department of Anesthesiology and Pain Medicine, University of Washington, Seattle, United States; [3]Department of Pharmacology, University of Washington, Seattle, United States

**Abstract** The parabrachial nucleus (PBN) is a major hub that receives sensory information from both internal and external environments. Specific populations of PBN neurons are involved in behaviors including food and water intake, nociceptive responses, breathing regulation, as well as learning and responding appropriately to threatening stimuli. However, it is unclear how many PBN neuron populations exist and how different behaviors may be encoded by unique signaling molecules or receptors. Here we provide a repository of data on the molecular identity, spatial location, and projection patterns of dozens of PBN neuron subclusters. Using single-cell RNA sequencing, we identified 21 subclusters of neurons in the PBN and neighboring regions. Multiplexed in situ hybridization showed many of these subclusters are enriched within specific PBN subregions with scattered cells in several other regions. We also provide detailed visualization of the axonal projections from 21 Cre-driver lines of mice. These results are all publicly available for download and provide a foundation for further interrogation of PBN functions and connections.

**\*For correspondence:**
jychen@uw.edu (JYC);
palmiter@uw.edu (RDP)

†These authors contributed equally to this work

## Editor's evaluation

The parabrachial nuclei are groups of neurons in the brainstem (one on each side) that integrate information about the state of the body to guide appropriate behavioral and homeostatic responses. The manuscript by Pauli and Chen et al. is a beautiful and much-needed study that characterizes the cell types that make up these nuclei. The result is a highly valuable resource to the academic community.

## Introduction

The parabrachial nucleus (PBN), located at the junction of the midbrain and pons, relays sensory information from the periphery primarily to the forebrain, thereby playing a major role in informing the brain of both the internal state (interoception) and external conditions (exteroception) to facilitate responses to adverse conditions and help maintain homeostasis.

The earliest reference to the PBN was from *Herrick, 1905*, who implicated it in transmitting gustatory signals and later elaborated upon by *Norgren and Leonard, 1971*. The inputs and outputs of the PBN have been studied extensively using anterograde and retrograde methods (*Fulwiler and Saper, 1984*; *Gauriau and Bernard, 2002*; *Krout and Loewy, 2000*; *Moga et al., 1990*; *Norgren, 1976*;

*Saper and Loewy, 1980*; *Tokita et al., 2009*). More recently, these studies have been supplemented using genetically engineered mice and stereotaxic delivery of viruses encoding fluorescent proteins to analyze the afferents and efferent projections of selected subsets of PBN neurons. The vagus transmits signals from internal organs, including the gastrointestinal system, to the nucleus tractus solitarius (NTS), which then projects to the PBN; thus, detection of visceral signals related to food and malaise depends on this circuit. Other internal organs, muscle, and bone transmit nociceptive signals via intermediary ganglia to the spinal cord and then directly to the PBN. Ascending fibers from the spinal cord relay peripheral temperature and pain signals directly to the PBN, while trigeminal neurons relay these signals from the face. In addition, blood-borne threats to homeostasis are detected by the area postrema and transmitted to the PBN (*Zhang et al., 2021*). Taste is transmitted from the tongue and palate via branches of the facial, petrosal, glossopharyngeal nerves to the rostral NTS and from there to the PBN. Additionally, calcium imaging and Fos-induction studies show that most sensory systems can activate neurons in the PBN (*Campos et al., 2018*; *Carter et al., 2013*; *Kang et al., 2022*), although the neuronal circuits involved are not well established. Thus, the PBN is a hub activated by a wide variety of sensory signals, which then report the state of the body to the other brain regions to elicit appropriate responses. Most of these afferent signals to the PBN are excitatory (glutamatergic), but there are also inhibitory, GABAergic inputs including those from the arcuate nucleus, bed nucleus of stria terminalis (BNST), and central nucleus of the amygdala (CEA). The PBN projects axons to the periaqueductal gray (PAG), extended amygdala (including the BNST, CEA, substantia innominata, SI), the cerebral cortex (primarily the insular cortex), thalamus, parasubthalamic nucleus (PSTN), hypothalamus, and medulla.

Pioneering neuroanatomical studies encouraged functional studies (lesions, pharmacological and viral/genetic interventions), which have substantiated the predictions that the PBN is important for responding to internal and external stimuli and maintaining homeostasis. Examples include taste, thermal sensation, visceral malaise, pain, itch, hypercapnia, breathing, cardiovascular control, arousal, hunger/satiety, thirst, sodium appetite, and alarm (*Palmiter, 2018*). The response of the PBN to these sensory modalities raises multiple questions: How many different neuron populations are there? Are specific neurons or subsets of neurons involved in transmitting each signal? Is there integration of sensory signals (crosstalk between neurons) within the PBN? Do individual neurons project axons to one target region or send collaterals to many brain regions? The location of neurons in the PBN that expresses distinct molecular markers will help address these questions.

The PBN is bisected by a fiber tract, the superior cerebellar peduncle (scp), resulting in subregions that are lateral or medial to the scp in primates. In rodents, the scp is rotated relative to that in primates such that so-called lateral regions are more dorsal to the scp, and medial regions are more ventral. Another fiber pathway, the ventral spinocerebellar tract (sctv), helps to define the dorsal border of the PBN in its caudal regions. The Kölliker-Fuse (KF) region (considered by some as part of the PBN), cuneiform nucleus, nucleus of the lateral lemniscus (NLL), mesencephalic trigeminal nucleus (MEV), and locus coeruleus (LC) are adjacent to it (*Dong, 2008*; *Fulwiler and Saper, 1984*; *Paxinos and Franklin, 2019*). In situ hybridization and immunohistochemistry studies have revealed that the PBN is primarily glutamatergic and expresses an abundance of different neuropeptides and neuropeptide receptors. These observations led to the creation of Cre-driver lines of transgenic mice that have been used to activate virally delivered, Cre-dependent genes to manipulate neuron activity and to visualize their axonal projections.

To define PBN cell types and gain insight into unique expression of signaling molecules and receptors, we adopted the single-cell RNA sequencing (scRNA-Seq) approach (*Hashikawa et al., 2020*; *Macosko et al., 2015*), which revealed 13 transcript-defined glutamatergic neuron types within the PBN proper along with sparsely interspersed GABAergic neurons. We then used in situ hybridization to anatomically locate the neurons within the PBN and Cre-driver lines of mice with viral expression of fluorescent proteins to establish the axonal projection patterns from the PBN of 21 Cre-driver lines of mice.

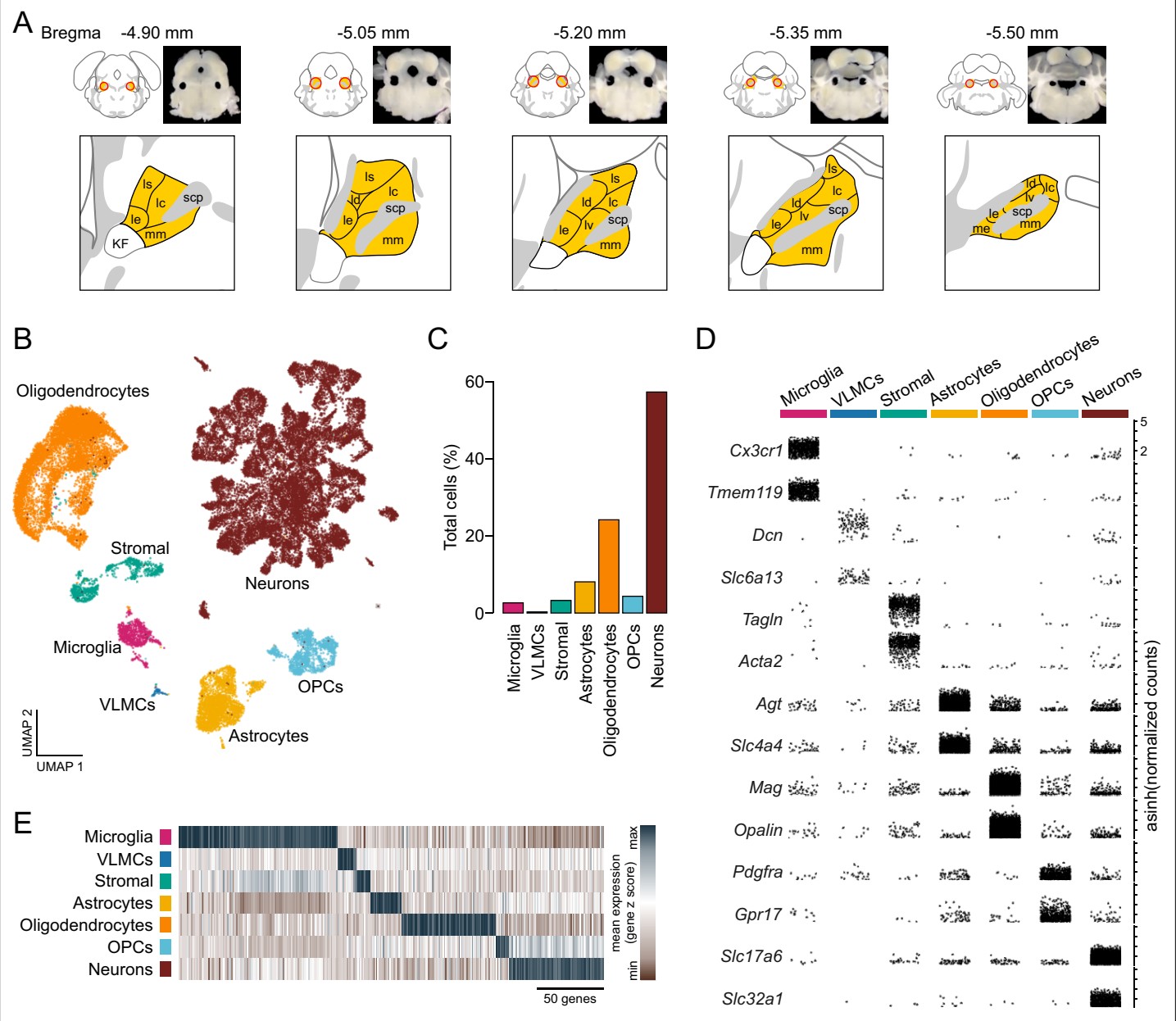

**Figure 1.** Single-cell RNA sequencing identifies resident cellular classes within the parabrachial nucleus (PBN). (**A**) Brain sections showing location of PBN and approximate boundaries of punches used for scRNA-Seq. PBN subregions from Allen Mouse Brain Atlas are shown in yellow; abbreviations are the same as in *Figure 4*. (**B**) Cells were clustered according to their transcriptional profiles and plotted in uniform manifold approximation and projection space. (**C**) Percentage of total cells comprised by each cluster. (**D**) Expression of canonical features across PBN clusters. Each point represents a single transcript plotted according to its asinh-normalized expression level. (**E**) Classes of PBN cell types are distinguished by unique transcriptional profiles comprised of multiple genes.

The online version of this article includes the following figure supplement(s) for figure 1:

**Figure supplement 1.** Technical metrics in scRNA sequencing analysis of resident parabrachial nucleus cell types.

## Results

### Single-cell RNA sequencing analysis of cell types in the PBN

To classify cell types in the mouse PBN according to their transcriptional profiles, we harvested brain tissue from 10 adult male and female C57BL/6J mice, excised the PBN along its rostral-caudal extension (*Figure 1A*), and prepared single-cell suspensions for high-throughput scRNA-Seq using a

commercial droplet-based assay (10× Genomics). After preprocessing the data to remove low-quality cells from the analysis (*Rossi et al., 2021*; *Stuart et al., 2019*), we retained a total of 39,649 single-cell barcodes that were sequenced to a median depth of 47,177 reads per cell. A total of 17,038 genes were detected, with a median of 1740 genes per cell (*Figure 1—figure supplement 1*).

To identify resident cell types of PBN tissue, cells were clustered on principal components and visualized in uniform manifold approximation and projection (UMAP) space (*Mcinnes et al., 2018*; *Figure 1B*). We then applied a likelihood ratio test to identify features that were differentially expressed between clusters (*Hafemeister and Satija, 2019*; *Macosko et al., 2015*; *McDavid et al., 2013*), and classified cells according to the specificity of canonical marker genes within each cluster (Materials and methods, *Figure 1B–E*, and *Supplementary file 1*). Low-resolution clustering identified seven transcriptionally distinct populations of neurons, glia, and stromal cells within the PBN (*Figure 1B–E*). Of these, neurons represented the largest proportion of cells at 57.2% (*Figure 1C*). Oligodendrocytes, marked by *Mag* and *Opalin*, were detected at 24.1% of total cells, and oligodendrocyte precursor cells (OPCs, 4.3% of total cells), were distinguished by their expression of *Pdgfra* and *Gpr17* (*Figure 1C–D*). The large number of oligodendrocytes and precursors is not surprising because the PBN is bisected by the scp. We detected a population of astrocytes representing 8.1% of cells that were labeled by robust expression of *Agt* and *Slc4a4*, as well as a smaller population of microglia (2.6%) that were specifically labeled by *Cx3cr1* and *Tmem119* (*Figure 1C–D*). Additionally, we identified two distinct populations of cells marked by stromal markers; one of these populations was characterized by its selective expression of *Tagln* and *Acta2* (3.2%), while another rare population of vascular leptomeningeal cells was marked by *Dcn* and *Slc6a13* (0.26% of total cells). Although known canonical markers were used for the biased identification of broad classes of resident cells within PBN tissue (*Figure 1B and D*), each of these cellular classes was marked by a robust profile of unique transcriptional features (*Figure 1E* and *Supplementary file 1*).

Next, we sought to identify distinct subclasses of neurons within the PBN. Subclasses of cells within a terminally differentiated cell-type share a similar transcriptional landscape, and as a result, statistically discriminating subclasses through clustering analysis require the presence of a small set of high-confidence, high-variance features. To enable high-resolution subclustering of PBN neurons, we first applied a more stringent quality threshold to the neuronal population isolated in the initial analysis (*Figure 2—figure supplement 1A–E*), resulting in a smaller set comprised 7635 neurons (*Figure 2A–B*). These cells were sequenced to a much higher median of 99,583 reads per cell with each cell containing a median of 3189 genes and 7823 transcripts (*Figure 2—figure supplement 1F–O*). We used a clustering approach like that applied for all cells and discriminated 21 unique subclusters of neurons (**N1-N21**) according to their expression of differential feature sets (Materials and methods, *Figure 2*, and *Figure 2—figure supplement 1P–Q*). We first classified neuronal subclusters as glutamatergic or GABAergic and designated **N1-N19** as glutamatergic with **N1, N2** being enriched in vesicular transporter *Slc17a7* (Vglut1), while **N3-N19** are enriched in vesicular transporter *Slc17a6* (Vglut2); the latter account for about 90% of all neurons sequenced (*Figure 2C–E*). The remaining two subclusters **N20, 21** are GABAergic based on expression of *Gad1, Gad2*, and *Slc32a1* (Vgat) (*Figure 2C–E*).

We sought to further define these 21 neuronal subclusters and identified two major clades that could be distinguished by their expression of transcription factors (*Figure 2F–G*). One major clade (**N4-N12**) is represented by expression of *Lhx2, Lhx9, Meis2*, and *Nrf1*. This clade probably descends from neurons that express *Atoh1* during development (*Karthik et al., 2022*). This group also includes nuclear receptors, *Nr2f1, Nr2f2, Nr4a*; zinc-finger protein, *Tshz1*; homeobox-containing transcription factors, *Barx2* and *Evx2*; and forkhead transcription factor, *Foxp2*. A subgroup of this clade is represented by *Nr4a2* and *Foxp2*, which are expressed prominently in five of the eight subclusters. The expression pattern of *Foxp2* in the PBN and its axonal projections via the ventral pathway (VP) to the hypothalamus has been described in detail (*Huang et al., 2021a*).

Another major clade that includes **N13-N18** (*Figure 2F–G*) is represented by expression of a group of homeobox-containing transcription factors *Lmx1a, Lmx1b, Pou2f2, Pou6f2, En1, Tlx3, Onecut2*, and *Satb2*, along with a member of the retinoic acid family (*Rorb*), the AP2 factor (*Tfap2b*), and co-repressor (*Tle4*). The two major clades represented by *Atoh1* (**N4-N12**) and *Lmx1* (**N13-N18**) descendants are largely non-overlapping populations (*Karthik et al., 2022*). They have distinct axonal

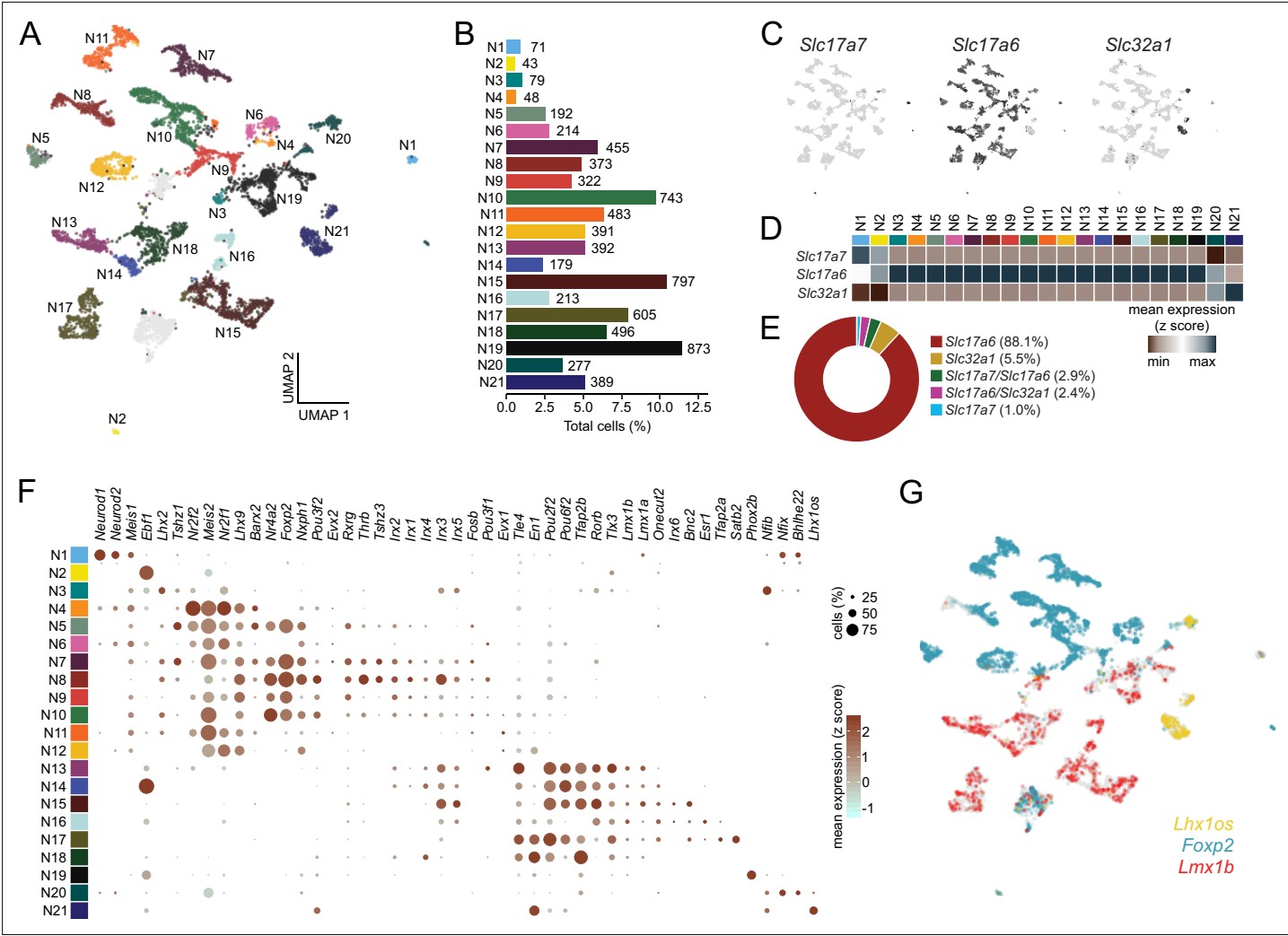

**Figure 2.** Single-cell RNA sequencing identifies discrete classes of parabrachial nucleus (PBN) neurons. (**A**) Neurons were clustered according to their transcriptional profiles and plotted in uniform manifold approximation and projection (UMAP) space. Two clusters were omitted from analysis (gray; see Materials and methods). (**B**) Percentage of total neurons comprised by each neuronal subcluster. (**C**) Expression values of fast neurotransmitters in UMAP space. (**D**) Average expression of fast neurotransmitters across neuronal subclusters. (**E**) Percentage of neurons individually expressing or co-expressing fast neurotransmitters. (**F**) Transcription factor expression across neuronal subclusters plotted according to their average normalized expression and fraction of cells expressing each gene. (**G**) Expression of the transcription factors *Foxp2*, *Lmx1b*, and *Lhx1os* across neurons in UMAP space.

The online version of this article includes the following figure supplement(s) for figure 2:

**Figure supplement 1.** Technical metrics in scRNA sequencing analysis of neuronal subclusters.

projection patterns, with the *Atoh1* clade following a VP to the forebrain and the *Lmx1* clade following the central tegmental tract (CTT).

The two neuron subclusters that express *Slc17a7* (**N1 and N2**) have distinct expression of transcription factors from each other and from the two major clades. Likewise, the two subclusters that express *Slc32a1* (**N20,** N**21**) also express distinct sets of transcription factors. As will be shown below, the two major clades (**N4-N12** and **N13-N18**) represent neurons that reside within the PBN. The remaining two subclusters contain cells that mostly border the PBN and express *Phox2b* (**N19**) or *Nfib* and *Lhx2* (**N3**) (**Figure 2F–G**).

Within these two major clades, we also found that the 21 neuronal subclusters could be further delineated based on preferential expression of specific genes (**Figure 3A**). Although our analysis identified multiple known subclasses of PBN neurons, we also identified novel neuronal subclusters and their corresponding transcriptional markers. Thus, this analysis provides a representation of PBN neuronal diversity at a higher resolution than has been previously appreciated (**Figure 3A–B**). A

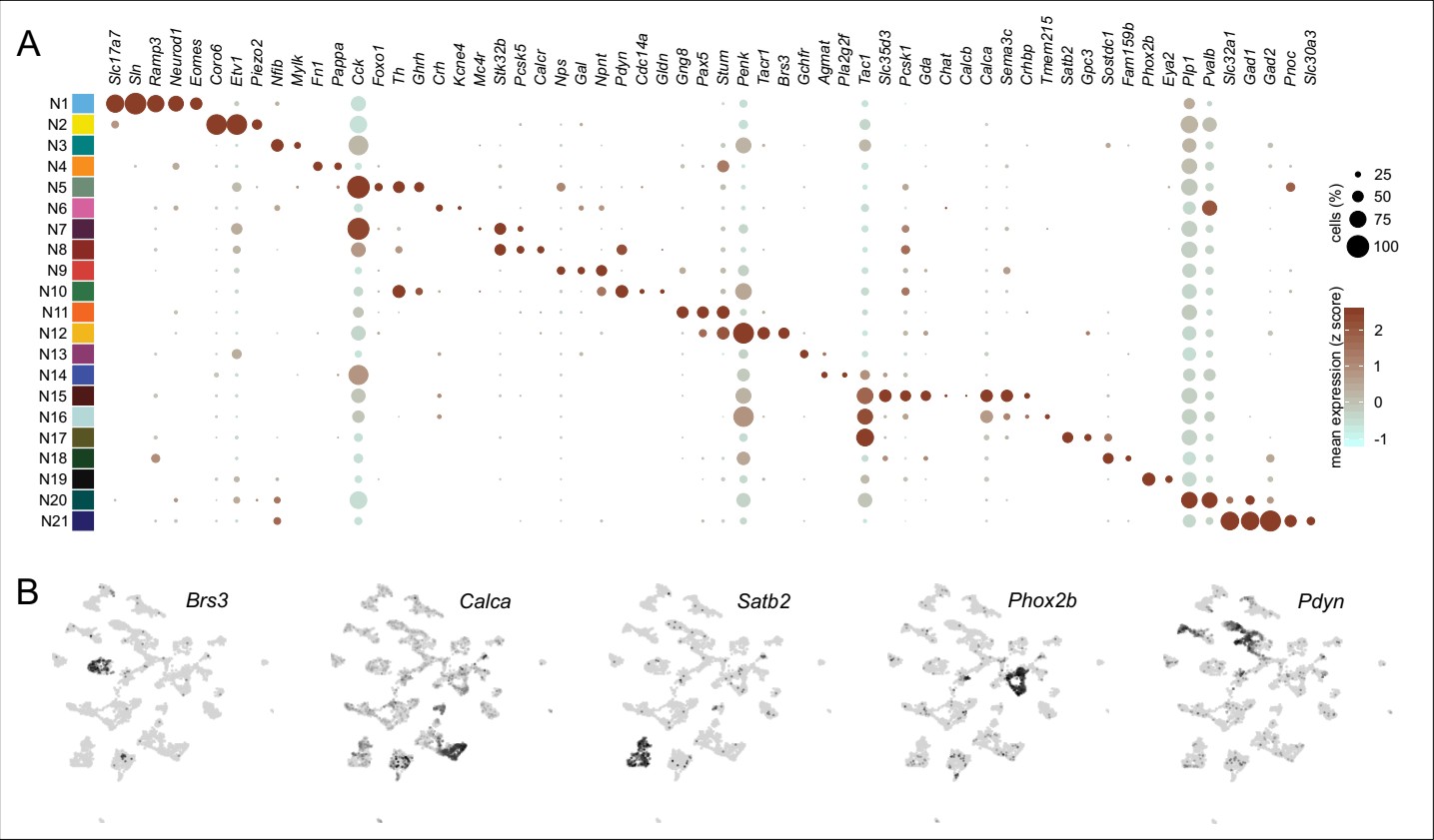

**Figure 3.** Distinguishing features of each neuronal subcluster. (**A**) Expression of select differentially expressed features across neuronal subclusters plotted according to their average normalized expression and fraction of cells expressing each gene. (**B**) Expression of select genes plotted in uniform manifold approximation and projection space.

summary file listing the average expression of each gene per cluster, fraction of cells expressing each gene within a cluster, and the differential expression p-value per cluster is provided for all genes in the dataset (*Supplementary file 2*).

## Expression of neuropeptides and G-protein-coupled receptors in the PBN

Neuropeptides are valuable markers for selected subpopulations of neurons in many brain regions. Some of these neuropeptide mRNAs are robustly expressed in one or two subclusters, e.g., *Calca, Ghrh, Nps, Npy, Pdyn, Pnoc*, while others are expressed in multiple subclusters, e.g., *Adcyap1, Cck, Crh, Gal, Gpr, Nmb, Nts, Penk, Tac1, Vgf* (*Table 1*, *Supplementary file 3*). Like the neuropeptide group, the genes encoding G-protein-coupled receptors (GPCRs) are genetically useful because they are likely to be expressed in neurons that receive aminergic or neuropeptide input; thus, making them desirable for circuit mapping (*Supplementary file 4*). Indeed, Cre-driver lines of mice have been generated for many neuropeptide and GPCR genes. GPCRs are also of interest because they are viable targets for drugs that could potentially modify the function of select neurons and circuitry in the PBN. The expression of GPCR genes is generally much lower than that of neuropeptides, making them more difficult to detect by immunohistochemistry or in situ hybridization. Consequently, scRNA-Seq data provide another resource for identifying the GPCRs expressed by distinct PBN neurons. GPCRs with relatively restricted expression are listed in *Table 1*.

## Locating neuronal populations by in situ hybridization

To determine if different neuron subclusters are in distinct PBN subdivisions, we performed fluorescent in situ hybridization on coronal sections of the PBN spanning the rostral-caudal axis from Bregma –5.0 to –5.4 mm. Representative probes for each neuronal subcluster (**N1-N21**) were chosen

**Table 1.** Neuropeptides and G-protein-coupled receptors (GPCRs) with restricted expression in the 21 neuronal subclusters.

These data are extracted from *Supplementary file 3* and *Supplementary file 4*.

| Subcluster | Neuropeptide | GPCR receptor |
|---|---|---|
| N1 | -- | -- |
| N2 | -- | *Adora2, Gpr156, Gpr157, Gprc5c[2], Olfr90, Olfr889,* **P2ry14\*** |
| N3 | *Apln* | *Cckbr[2], Mc4r[3], Mchr1[2]* |
| N4 | **Npy[2]** | *Fzd2, Grpr, Ntsr1[2]* |
| N5 | *Ghrh[2], Nmb[2],* **Nps\*[2]**, **Pnoc\*[2]** | *Olfr552, Qrfpr* |
| N6 | **Crh\*** | *Lgr5* |
| N7 | **Grp\*[2]**, *Npy[2], Prok2[3]* | *Chrm1[2], Fzd7, Fzd8[2], Mc4r[3], Npbwr1[2], Npy2r, Rxfp3* |
| N8 | **Grp\*[2]**, **Pdyn\*[2]**, *Prok2[3]* | **Calcr[3]**, *Cckbr[2], Ednra[2], Fzd8[2], Gpr6, Mc3r, Npbwr1[2]* |
| N9 | **Gal\***, **Nps\*[2]** | *Chrm1[2], Ednra[2], Mchr1[2]* |
| N10 | *Ghrh[2], Nmb[2],* **Penk\*[3]**, **Pdyn\*[2]**, *Prok2[3]* | *Agtr2, Mc4r[3]* |
| N11 | *Edn1, Sct* | *Calcr[3], Ntsr1, Olfr876* |
| N12 | **Penk\*[3]** | **Brs3**, *Calcr[3],* **Tacr1\*** |
| N13 | *Nmb[2], Nmu* | *Ptgfr* |
| N14 | -- | *F2rl2[2], Gabrb2[2], Hrh2[3]* |
| N15 | **Calca\***, *Calcb, Gast,* **Tac1\*[3]** | *Avpr1a, F2rl2[2], Galr1, Hrh2[3], Npr3[4]* |
| N16 | **Calca\***, **Penk\*[3]**, **Nts\***, **Tac1\*[3]** | *Cckar, F2rl2[2], Hrh2[3], Npr3[4]* |
| N17 | **Tac1\*[3]** | *Gabrb2[2],* **Npr3[4]** |
| N18 | -- | *Npr3[4]* |
| N19 | -- | -- |
| N20 | -- | *Gprc5c[2]* |
| N21 | **Pnoc\*[2]**, *Trh* | -- |

Bold, highly expressed and unique to this subcluster.
Bold\*, highly expressed in this subcluster but also expressed in others at lower levels.
Superscript, number of subclusters with expression.

based on the distinguishing genes in each subcluster (*Figure 4A*). Two sets of 12 probes were used in HiPlex experiments and a composite image locating each subcluster within the PBN was generated based on the Allen Mouse Brain Atlas (AMBA) (*Dong, 2008*; *Lein et al., 2007*) as a guide (*Figure 4B*). The tissue punch for scRNA-Seq was centered on the PBN, but neighboring regions were also included; consequently, only 13 glutamatergic of the 19 glutamatergic UMAP subclusters are within the PBN (*Figure 4C*). An example of the results obtained with three probes (*Calca, Brs3,* and *Phox2b*) is shown in *Figure 4—figure supplement 1*. PowerPoint summaries and individual TIFF images of all the HiPlex probes for the 5 Bregma levels of the PBN examined are available at Zenodo (DOI: 10.5281/zenodo.6707404). Qualitative scoring of the expression of each probe within a specific PBN subregion (*Figure 5A*) or neighboring regions (*Figure 5—figure supplements 1 and 2*) is based on the number of transcripts (signal intensity) and number of cells (*Figure 5B*). The HiPlex data were supplemented with RNAScope experiments to further investigate differential gene expression within specific subclusters.

The subregions of the PBN were derived from cytoarchitectural characteristics based on Nissl staining of rodent tissue sections and supplemented with anterograde and retrograde labeling approaches. We referenced the AMBA because it is widely used and includes a wealth of in situ hybridization and connectivity data, which was useful as a guide for determining the general location

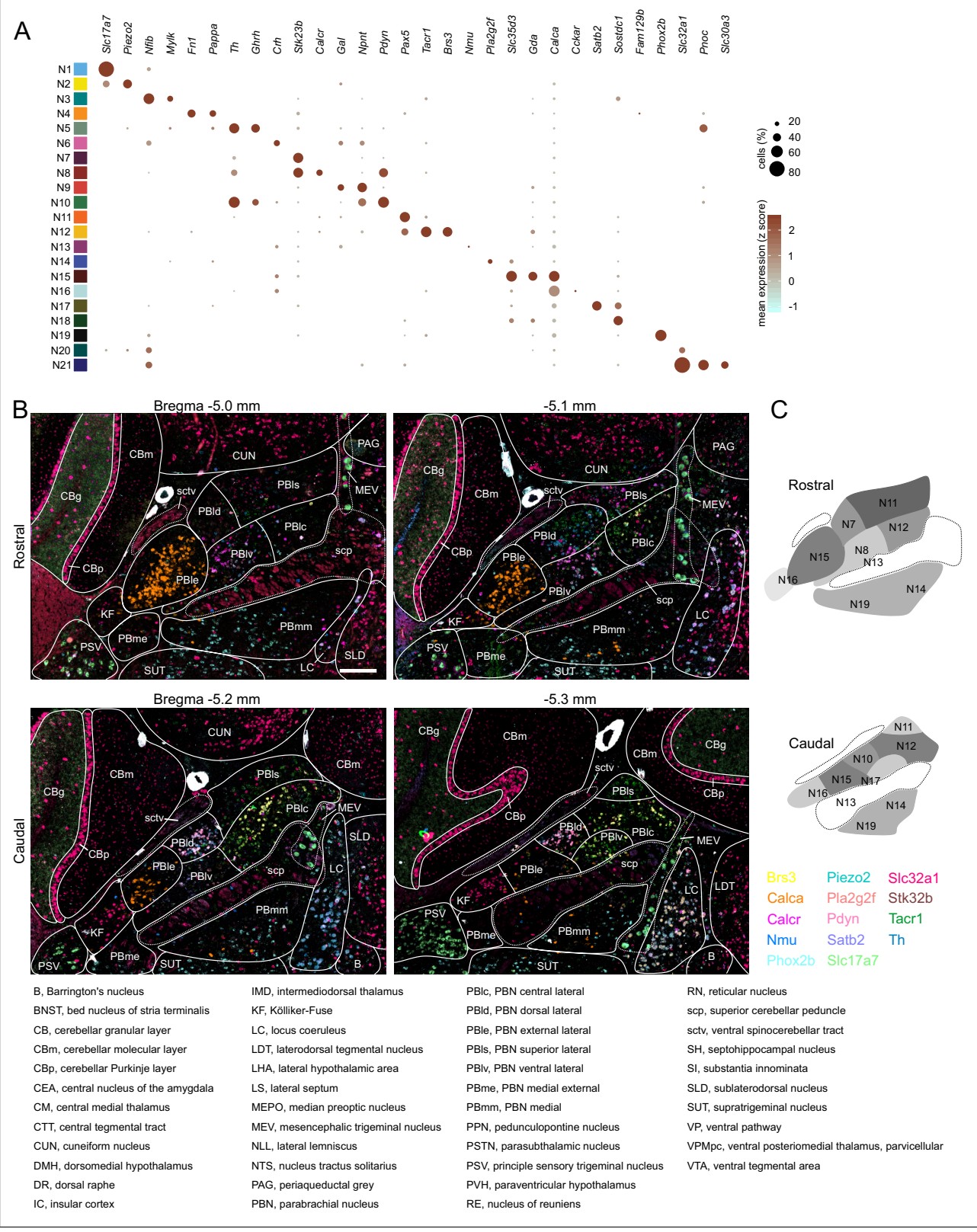

**Figure 4.** Localization of mRNAs in subregions of the parabrachial nucleus (PBN) based on HiPlex results. (**A**) Expression of genes selected as HiPlex probes within the scRNA-seq dataset. (**B**) Example of how the regions of interest denoting PBN subregions were drawn for analysis of 15 probes from the first HiPlex experiment. Probes and their colors are indicated. (**C**) Diagram of the approximate location of 12 of the identified clusters in a rostral and caudal PBN. Scale bar, 200 μm. (**D**) List of abbreviations from AMBA used throughout the manuscript and figures.

*Figure 4 continued on next page*

Figure 4 continued

The online version of this article includes the following figure supplement(s) for figure 4:

**Figure supplement 1.** Example of HiPlex staining for *Brs3*, *Calca*, and *Phox2b* for five Bregma levels.

PBN subregions. The nomenclature in the AMBA is not entirely consistent with the atlas (*Paxinos and Franklin, 2019*) nor that developed for the rat (*Fulwiler and Saper, 1984*). For example, the internal lateral PBN does not exist in the AMBA, (it is called the superior lateral region, PBls), it does not divide the external lateral region into 'inner' and 'outer' regions but includes a lateral ventral region instead, and it does not show the lateral crescent. The boundaries of the PBN extend beyond those shown in the AMBA; neurons with PBN characteristics reside in regions that extend rostral to Bregma -5.0 and dorsally into what the AMBA indicates is the pedunculopontine nucleus (PPN) and the NLL (as

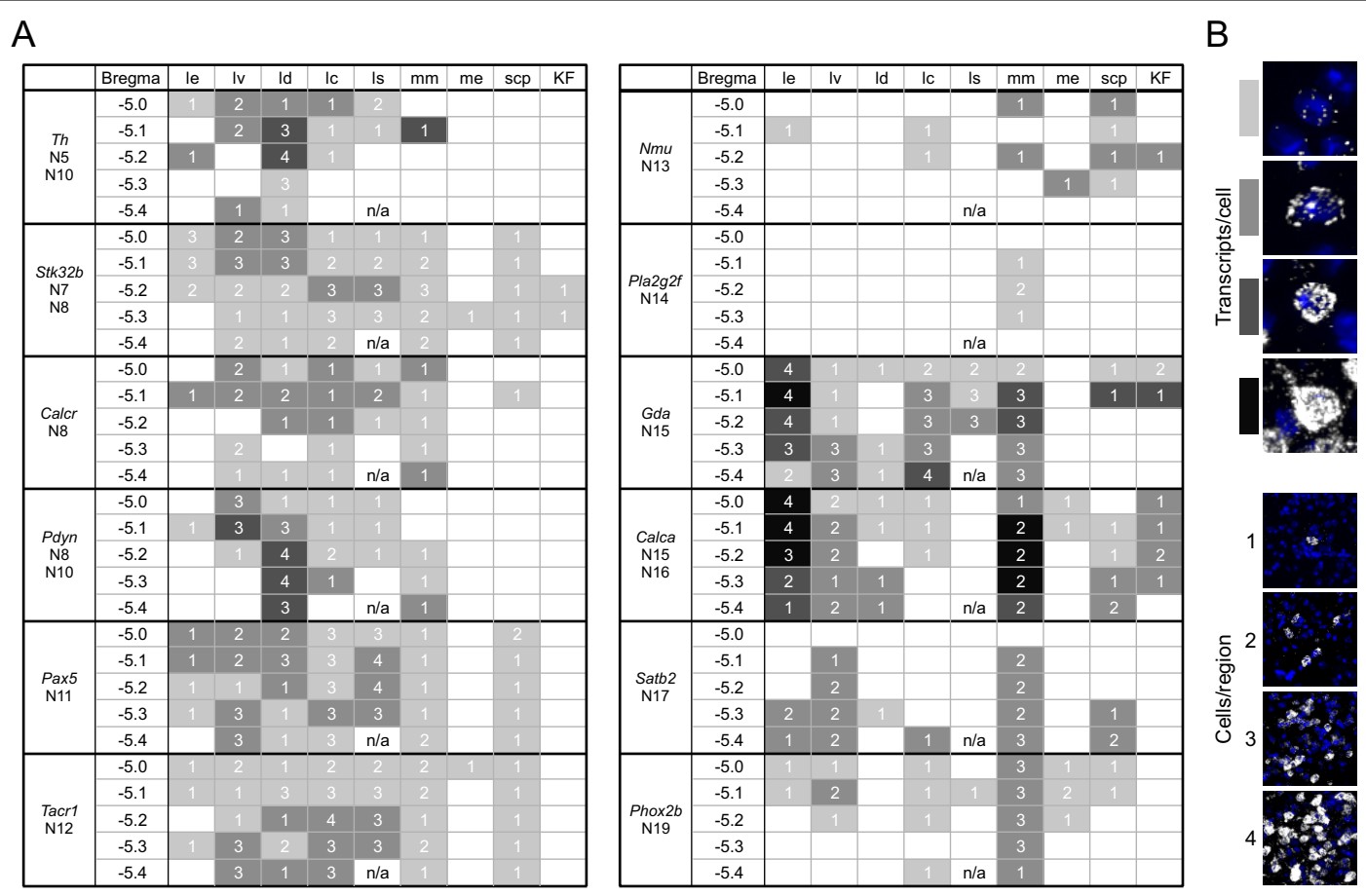

**Figure 5.** A guide to HiPlex results showing relative strength and abundance of mRNA expression in parabrachial nucleus (PBN) subregions. (**A**) Qualitative expression for 12 genes that can be used to identify different subregions in the PBN. Strength of expression and percentage of cells in subregions of the cells were analyzed for five bregma levels. (**B**) Key for the colors and numbers in the table. Shade of gray gets darker as the number of transcripts per cell increases, and the number represents an estimate of the number of positive cells per subregion. The abbreviations are defined in *Figure 4*.

The online version of this article includes the following figure supplement(s) for figure 5:

**Figure supplement 1.** Guide of HiPlex results for all probes with signal in parabrachial nucleus and surrounding area, page 1.

**Figure supplement 2.** Guide of HiPlex results for all probes with signal in parabrachial nucleus and surrounding area, page 2.

**Figure supplement 3.** RNAscope for *Cck*, *Gal*, *Nps*, *Ntsr1*, and *Th* in far rostral parabrachial nucleus (PBN).

**Figure supplement 4.** RNAscope for *Calca* and *Gda*.

**Figure supplement 5.** RNAscope for *Calca* and *Slc17a6*.

**Figure supplement 6.** RNAscope for *Calca*, *Satb2*, and *Tac1*.

described later). There is also lack of agreement on whether the KF is part of the PBN. It is unfortunate that the order of letters in the abbreviations used for subregion of the PBN does not follow common usage, e.g., the external lateral PBN is abbreviated PBle in AMBA and LPBE in Paxinos and Franklin. All atlases were generated without consideration of cellular phenotype (protein and gene expression); thus, we did not expect that neuronal subclusters identified in this study would fit nicely within atlas boundaries, or that they are the defining characteristics of those subregions. For all of the probes shown in *Figure 5A*, the general theme is that genes are predominantly expressed in one location with scattered expression in several other subregions.

Before discussing the glutamatergic subclusters within PBN, we first comment on distinguishing feature probes that were not, or rarely, detected in the PBN. Clusters **N1** and **N2** express *Slc17a7* encoding Vglut1. While we detected a few *Slc17a1*-expressing cells in the lateral PBN (*Figure 5— figure supplements 1 and 2*), consistent with the AMBA, the majority of **N1** cells are likely due to contamination from the cerebellar granular layer included in the tissue punch. **N2** represents the MEV that expresses both *Slc17a7* and *Piezo2*. We did not detect strong signals within the PBN for clusters **N3** (probes *Mylk, Nfib*), **N6** (*Crh*), or **N18** (*Shisal2b, Sostdc1*). Most of these probes labeled cells outside the PBN (*Figure 5—figure supplement 1*).

## N4

*Fn1* and *Pappa* were originally chosen to represent **N4**, but the transcripts were either exclusively in vasculature (*Fn1*) or too weak and dispersed to see obvious positive cells (*Pappa*). Our HiPlex experiment (Bregma 5.0–5.4 mm) did not include rostral PBN-associated cells that are labeled as PPN and NLL by the AMBA (*Huang et al., 2022*). Thus, we examined expression of *Ntsr1* and *Cck* as representatives of **N4** by RNAScope in rostral sections (Bregma -4.8 mm). *Ntsr1*-expressing neurons are a subset of a much larger *Cck* population along the border of the PPN and NLL shown in the AMBA (*Figure 5—figure supplement 3*). This *Ntsr1* population was confirmed by viral tracing using the *Ntsr1^Cre* line (see next section). These cells extend caudally and beyond the sctv fiber tract that AMBA shows as a PBN boundary. While *Cck* is widely expressed in many subclusters, this rostral cluster of *Ntsr1*-expressing neurons corresponds to what *Fulwiler and Saper, 1984* referred to as PBls.

## N5, N6

*Th*, *Ghrh*, *Pnoc,* and *Nps* were chosen to represent **N5,** but we did not find any cells that co-expressed these mRNAs in the PBN even though each of the individual probes detected cells in the PBN. *Crh* was chosen to represent **N6**, but aside from weak co-expression with *Calca* in **N15/N16**, most of the *Crh*-expressing neurons were located outside the PBN in Barrington's nucleus.

## N7

*Stk32b* was chosen to localize **N7**. It is expressed widely throughout the PBN and neighboring brain regions. Its expression also overlaps with other markers, e.g., some *Calca* and *Pdyn* neurons; however, there is a *Stk32b* cluster in rostral PBld that does not overlap with other markers that could represent **N7**.

## N8, N10

scRNA-Seq data suggested that *Pdyn* would be co-expressed with *Stk32b* and *Calcr* in **N8**, which was confirmed by HiPlex. This subcluster is closer to the scp in the rostral PBlv region. In addition, **N10** is represented by *Pdyn, Ghrh, Pnoc, Crh,* and *Th*, which were also confirmed by HiPlex; these neurons are in more caudal PBld regions. Although *Th* mRNA is expressed in many *Pdyn* neurons, it is unlikely that they are catecholaminergic because TH protein levels are very low compared to that in LC and expression of other mRNAs that are required for catecholamine synthesis, vesicular transport, and re-uptake were not detected. There is also an intriguing population of *Th* neurons that do not express *Pdyn* in the PBld. Transduction of *Pdyn^Cre* mice with AAV carrying Cre-dependent effector genes has been used to assess the role of *Pdyn*-expressing neurons, which include temperature regulation, nocifensive behaviors, and feeding (*Chiang et al., 2020*; *Geerling et al., 2016*; *Kim et al., 2020*; *Norris et al., 2021*; *Yang et al., 2021*). It will be important to learn whether the two clusters of *Pdyn* neurons have distinct functions.

### N9

*Npnt* cells without *Pdyn* were originally chosen to locate **N9**, but the few cells of this type often express *Crh* and are not part of the PBN. **N9** was also predicted to express *Gal* and *Nps*. Because *Nps* is in the far rostral region of PBN (*Huang et al., 2022*), we used RNAscope to show *Nps*-positive cells that co-express *Gal* reside ventral to the **N4** population of *Ntsr1/Cck* neurons along the border PPN and NLL shown in AMBA. There is also a small population of *Th*-positive neurons medial to the *Nps/Gal* cells (*Figure 5—figure supplement 3*).

### N11, N12

The transcription factor *Pax5* was chosen as a marker for **N11** and **N12**, though the *Pax5* signal is more diffuse than other HiPlex probes. There is a distinct group of *Pax5*-positive, *Pdyn*-negative cells in the rostral PBlv close to scp that could represent **N11**. *Brs3* (*Figure 4—figure supplement 1*) and *Tacr1* are co-expressed and represent a subset of *Pax5* neurons in rostral PBlc and PBls as **N12**. Expression of *Tacr1* and *Pax5* is more widespread than *Brs3*, extending into PBls and PBmm in caudal sections. Studies using *Tacr1Cre* mice revealed that they play important roles in coping behaviors after painful events (*Barik et al., 2021*; *Deng et al., 2020*; *Huang et al., 2019*; *Ma, 2022*).

### N13, N14

*Nmu* was chosen for **N13**. A small number of *Nmu*-expressing neurons were observed in mid-PBN sections that are scattered within the scp or adjacent PBmm, in agreement with location of cell bodies revealed by expression of a fluorescent protein in a *NmuCre* line of mice that were bred with a *Rosa26* reporter line (Jarvie and Knight, personal communication). *Pla2g2f* was chosen to locate **N14**; there is a small cluster of weakly expressing cells in the PBmm of mid-PBN sections. The transcription factor *Ebf1* is also robustly expressed in **N14** (*Figure 2B and F*), though we did not investigate its expression using in situ hybridization.

### N15, N16

*Calca* is primarily expressed in the PBle as revealed by AMBA, our Hiplex analysis, immunohistochemistry (*Shimada et al., 1985*), and viral expression of reporter genes activated by injection into the PBN of *CalcaCre* mice (*Bowen et al., 2020*; *Carter et al., 2013*; *Chen et al., 2018*; *Huang et al., 2021b*, *Kaur et al., 2017*). *CalcaCre* mice have been used extensively to examine the role of these neurons in nocifensive behaviors, appetite, and arousal (*Palmiter, 2018*). Many *Calca*-expressing cells are also scattered throughout the lateral and medial PBN, which are especially prominent in rostral sections (*Figure 4—figure supplement 1*). *Calca* expression extends into the caudal KF region where it is intermingled with *Slc32a1*-expressing neurons. *Calca* is also expressed in the LC based on in situ hybridization results in agreement with *Huang et al., 2021b*; the LC was deliberately excluded when making the tissue punches for scRNA-Seq. It was unexpected that the scRNA-Seq analysis would reveal *Calca* expression in two subclusters; **N15** neurons are threefold more abundant than **N16** neurons (*Figure 2B*). To distinguish between the two clusters, we used RNAScope with probes for *Calca* and *Gda* (**N15** enriched). *Calca* cells with strong expression in PBle almost entirely overlap with *Gda*, along with both strongly and weakly expressing *Calca* cells in other subregions (**N15**). There is a small group of *Calca* cells without *Gda* in the lateral ventral PBle and partially into the KF region that likely represent **N16**, as well as some cells scattered sparsely throughout the PBN (*Figure 5—figure supplement 4*). scRNA-Seq data reveal that *Calca* and *Chat* are co-expressed in **N15** but not **N16**, while *Calca* and *Esr1* are co-expressed in **N16** but not **N15**. Note that the relative abundance of transcription factors expressed in these two *Calca* subclusters is also different (*Figure 2F*). Distinguishing between the axonal projections and functions of these two *Calca* clusters will be informative.

RNAScope experiments with *Calca* and *Slc17a6* probes reveal that most of the *Calca*-expressing neurons within PBle form a tight cluster with few *Slc17a6*-only neurons interspersed. This arrangement is even more apparent when *Slc17a6* expression was inactivated using a virus expressing SaCas9 and two guide RNAs targeted to inactivate *Slc17a6* (*Hunker et al., 2020*); in these experiments, *Slc17a6* signal was uniformly weak in the center of the *Calca*-expressing cells (*Figure 5—figure supplement 5*). Note that there are *Slc17a6*-expressing cells without *Calca* that reside between the *Calca* cluster and the scp. Some authors distinguish between PBle 'outer' where *Calca* neurons reside and 'inner'

**Table 2.** Enrichment of mRNAs for neuropeptides in *Calca* neurons based on RiboTag experiment. Enrichment is measured as the ratio of immunoprecipitated (Ippt) to input. Housekeeping and glial mRNAs are included for reference.

| Gene | Input | Ippt | Enrichment | |
|------|-------|------|------------|---|
| *Calca* | 373 | 2277 | 6.10 | |
| *Nts* | 175 | 1013 | 5.78 | |
| *Vgf* | 300 | 1343 | 4.47 | |
| *Cbln2* | 266 | 1138 | 4.27 | |
| *Adcyap1* | 923 | 3299 | 3.57 | |
| *Tac1* | 243 | 934 | 3.84 | |
| *Sst* | 1011 | 2583 | 2.55 | |
| *Scg2* | 3499 | 7641 | 2.18 | |
| *Nucb2* | 266 | 563 | 2.11 | |
| *Pnoc* | 264 | 549 | 2.07 | |
| *Chga* | 825 | 1597 | 1.93 | |
| *Cartpt* | 115 | 189 | 1.64 | |
| **Reference genes** | | | | |
| *Gapdh* | 3192 | 3583 | 1.12 | Housekeeping |
| *Actb (5)** | 8340 | 6640 | 0.80 | Housekeeping |
| *Gfap* | 1219 | 532 | 0.44 | Astrocytes |
| *Mbp (2)* | 9395 | 4148 | 0.44 | Oligodendrocytes |
| *Aif1 (3)* | 1914 | 512 | 0.27 | Microglia |
| *S100b* | 1330 | 331 | 0.25 | Oligodendrocytes/astrocytes |

*(...) average of n values.

where these *Calca*-negative cells reside; in the AMBA this region is referred to as PBlv. Some of these cells express *Tac1* (**Figure 5—figure supplement 6**). There are few, if any, GABAergic neurons in PBle (compare *Calca* and *Slc32a1* in **Supplementary file 5**).

Both the *Calca* and *Calcb* genes encode calcitonin gene-related peptide (CGRP), but *Calcb* is only weakly expressed based on scRNA-Seq, which is consistent with the observation that *Calca*-null mice express negligible CGRP immunofluorescence in the PBN (**Allen et al., 2022**; **Chen et al., 2018**; **Zajdel et al., 2021**). The AMBA shows robust expression of *Calcb* perhaps because their probe hybridized to both genes.

The co-expression of neuropeptides along with CGRP was verified by immunoprecipitation of polysomes from *Calca^Cre* mice expressing *Rpl22^HA* (RiboTag) followed by microarray analysis of mRNAs (**Sanz et al., 2009**), which revealed enrichment of mRNAs for *Nts, Tac1, Adcyap1,Vgf,* and several more (**Table 2**); these results were confirmed by scRNA-Seq (**Supplementary file 3**). The enrichment of these neuropeptide mRNAs was lower than *Calca*, suggesting that they are expressed in subsets of *Calca* neurons. Consequently, there could be a complex mixture of cells expressing various combinations of these neuropeptides. Expression of two GPCR mRNAs (*Avpr1a* and *Galr1*) is restricted to **N15,** and *Cckar* is restricted to **N16** (**Table 1**). Several other GPCRs, including *Oprm1* which plays an important role in opioid-induced respiratory depression (**Liu et al., 2022**), are expressed along with *Calca*, but also in other clusters (**Supplementary file 4**). *Chat* (encodes acetylcholine biosynthetic enzyme) is expressed in **N15** (as well as **N6**), a result consistent with the location of fluorescent protein expression after viral transduction of *Chat^Cre* mice with AAV-DIO-YFP and immunohistochemistry (**Garfield et al., 2015**; **Nasirova et al., 2020**). *Slc18a3* mRNA, which encodes the vesicular transporter

for acetylcholine, is selectively expressed in **N15**, suggesting that this *Calca* neuron population is both glutamatergic and cholinergic.

### N17

*Satb2*, which encodes a transcription factor, is a defining gene for this subcluster. These neurons are scattered in the PBlv, and PBmm and scp in the caudal PBN based on HiPlex results and reporter gene expression from *Satb2^Cre* mice. scRNA-Seq experiments predicted that *Satb2* neurons co-express *Tac1*, which was confirmed by an RNAScope experiment in which *Tac1, Satb2,* and *Calca* probes were combined (*Figure 5—figure supplement 6*). This experiment revealed co-expression of *Tac1* and *Calca* as well as *Tac1* and *Satb2*, but there were also abundant *Tac1* cells without expression of either *Satb2* or *Calca*, especially in PBlv and PBlc. *Satb2^Cre* mice have been used to show that these PBN neurons relay taste signals to the thalamus (*Fu et al., 2019*; *Jarvie et al., 2021*).

### N18

Neither the *Shisal2b* nor *Sostdc1* probes chosen for this cluster gave a definitive signal in the PBN.

### N19

The transcription factor *Phox2b* was chosen as a distinct marker for **N19**. There is a large population of *Phox2b*-expressing neurons in the supratrigeminal area below the PBN, but some of these neurons reside in the PBmm and near the scp (*Figure 4—figure supplement 1*). *Phox2b* is also expressed in LC and KF, in agreement with *Karthik et al., 2022*.

### N20, N21

*Slc32a1*, *Gad1* and *Gad2*, are expressed in neurons scattered throughout the PBN, with some small clusters of *Slc32a1*-positive neurons in PBls, PBmm, and caudal KF. GABAergic neurons in PBN region can inhibit local glutamatergic neurons (*Sun et al., 2020*). scRNA-Seq indicated that *Pnoc* is expressed in **N21** but not **N20**. Most *Slc32a1*-expressing cells in caudal KF region do not express *Pnoc*. Some GABAergic cells scattered throughout the rest of the PBN express *Pnoc* while others do not; thus, **N20** and **N21** appear to be intermingled throughout the PBN and KF.

In summary, of the neuron subclusters identified by scRNA-Seq, some are in brain regions adjacent to the PBN (**N1-3),** some could not be identified (**N5, 6, and 18**), and the GABAergic cells are sparsely scattered throughout the PBN (**N20, 21**). That leaves 13 distinct clusters of excitatory neurons (**N4, 7–17, and 19**) that are expressed within the PBN (*Figure 4C*, *Figure 5—figure supplement 1* and **2**). Each of these PBN subclusters typically has prominent expression of the distinguishing genes in one subregion with less frequent expression in other subregions.

## Mapping PBN expression and axonal projections with Cre-driver lines of mice

Expression of fluorescent proteins from AAV injected into the PBN of Cre-driver lines of mice provides an independent means of locating the cell bodies and visualizing the axonal projections. We injected AAV1-Ef1a-DIO-YFP and AAV1-Ef1a-DIO-synaptophysin:mCherry into the PBN of 21 Cre-driver lines to visualize cell bodies and processes within the PBN (YFP) and synapses (mCherry) throughout the entire brain. The viruses were bilaterally injected into PBN of four to five mice; after at least 3 weeks for viral expression, the mice were perfused for histology, and a preliminary analysis of viral expression in the PBN region was performed. Brains with the most precise viral placement were used to generate the TIFF stacks of the entire brain, available on Zenodo (DOI: 10.5281/zenodo.6707404). The Cre-expressing cells in the PBN were categorized by their location within large PBN subdivisions and surrounding regions (*Figure 6A–E*). In addition to the 21 Cre-driver lines described here, PBN expression from additional Cre-driver lines has been reported (*Table 3*), not including those described in the Allen Institute Connectome project. *Slc17a6^Cre* (Vglut2), which is expressed in most of the subclusters, reveals the overall distribution of glutamatergic projections from the PBN (*Huang et al., 2019*). As expected, the location(s) of fluorescent cell bodies in the PBN of the Cre-driver lines

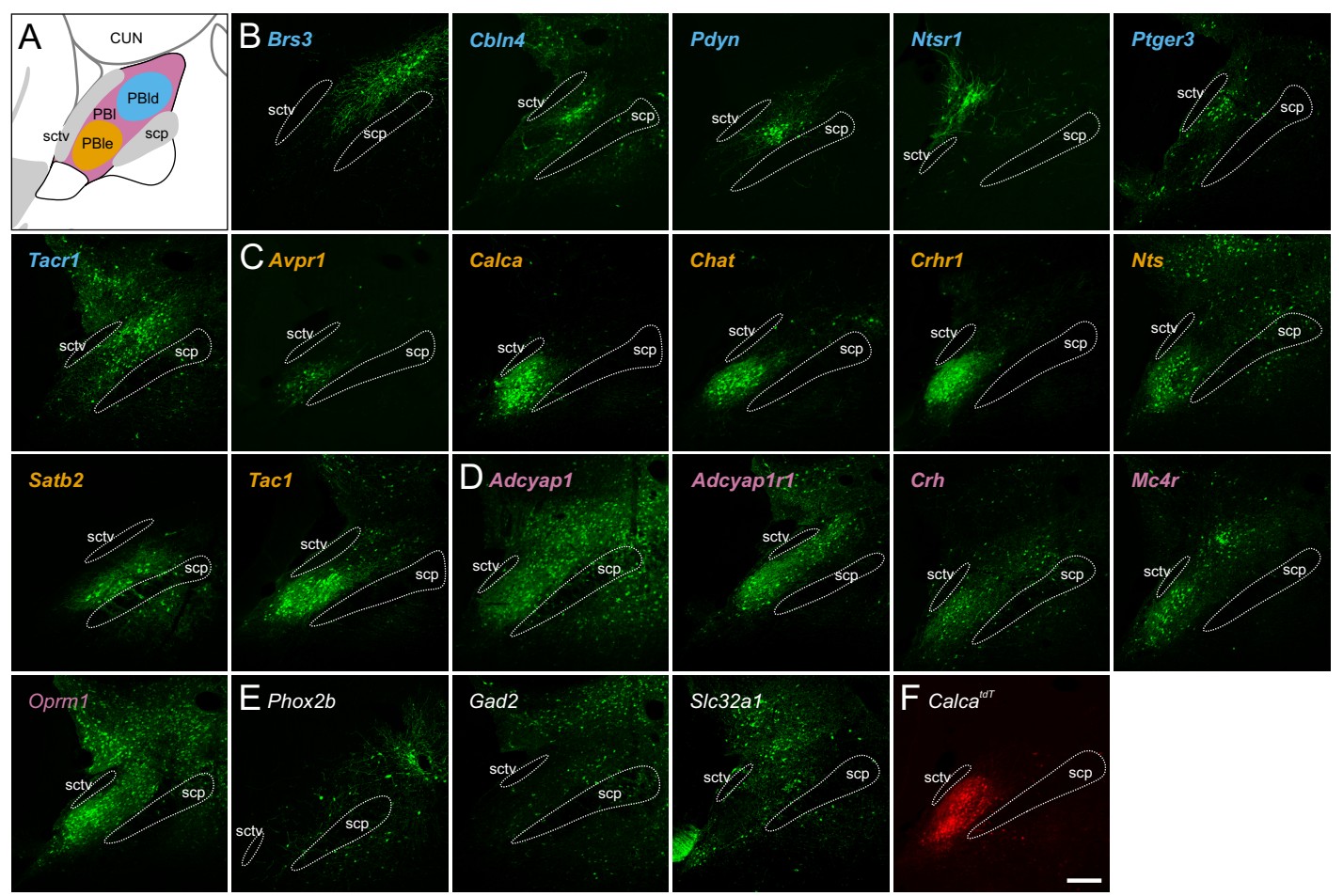

**Figure 6.** Parabrachial nucleus (PBN) expression in Cre-driver mouse lines. (**A**) Schematic of the PBN showing PBle in orange-dotted line, dorsal PBN regions in blue-dotted line, and expression in both in pink. (**B**) Five Cre-driver lines (blue lettering) with expression primarily in dorsal PBN. (**C**) Six Cre-driver lines (orange) primarily in PBle. *Satb2* is included here because its projection pattern resembles that of this group. (**D**) Five Cre-driver lines (pink) with expression in several PBN regions. (**E**) Five Cre-driver lines (gray) with expression patterns that do not fit with the other categories. (**F**) Image of *Calca-tdTomato* expression in the PBN for comparison. All images of viral expression are in mid-PBN sections; approximately Bregma –5.2 mm. Scale bar, 200 μm. Source data available at Zenodo DOI: 10.5281/zenodo.6707404 and includes complete TIFF stacks for each of these Cre-drivers and *Calca^tdT^*.

of mice is consistent with in situ hybridization results, although the promoter in the virus and multiple viral particles per cell can provide more robust expression than the endogenous gene, e.g., many GPCRs where the in situ signal in the AMBA is undetectable.

We also mapped the expression of *Calca* in mice with tdTomato targeted to the *Calca* locus (*Calca^tdT^*, *Figure 6F*). Fluorescence from *Calca^tdT^* mice reveals that in addition to robust expression in PBle, *Calca*-expressing cells are scattered throughout the lateral and medial PBN, which are especially prominent in rostral sections in agreement with the in situ experiments above. This approach also reveals tdTomato fluorescence in the caudal KF, LC, trigeminal, and facial nuclei, in agreement with the AMBA and (*Huang et al., 2021b*). Note that a genetic cross of *Calca^Cre^* mice with Cre-dependent reporter line, e.g., Gt(*ROSA*)26Sor^lsl-tdTomato^ (Ai14) results in widespread fluorescence throughout the brain, presumably due to developmental expression (*Carter et al., 2013*).

For each of the 21 Cre-driver lines and the *Calca^tdT^* line that we analyzed, images were taken from every third coronal, 35-μm section, stitched together and registered to generate a TIFF stack that can be manipulated to view expression of YFP and mCherry throughout the brain using ImageJ. All images are available to view and download from Zenodo; an example of *Calca* neuron expression is shown in *Video 1*.

**Table 3.** Cre-driver lines of mice used in parabrachial nucleus (PBN) studies.
List of mice (and source) that have been used to study PBN expression and projection patterns.

| Cre-driver | Extent of analysis | Source | Identifiers |
|---|---|---|---|
| *Adcyap1* | Extensive | This paper | JAX Strain #:030155 |
| *Adcyap1r1* | Extensive | This paper | JAX Strain #:035572 |
| *Avpr1a* | Extensive | This paper | JAX Strain #:035573 |
| *Brs3* | Moderate to extensive | This paper; *Mogul et al., 2021* | JAX Strain #:030540 |
| *Calca* | Limited to extensive | This paper; *Chen et al., 2018*; *Bowen et al., 2020*; *Huang et al., 2021b*; *Kaur et al., 2017* | JAX Strain #:033168 |
| *Cbln4* | Extensive | This paper (previously unpublished; see Methods) | |
| *Cck* | Limited to extensive | *Grady et al., 2020*; *Yang et al., 2020* | JAX Strain #: 012706 |
| *Chat* | Extensive | This paper | JAX Strain #: 006410, 031661 |
| *Crh* | Extensive | This paper | JAX Strain #: 012704 |
| *Crhr1* | Extensive | This paper, *Sanford et al., 2017* | |
| *Esr1* | Limited | *Grady et al., 2020* | JAX Strain #: 017913, 031386 |
| *Gad2* | Moderate | This paper | JAX Strain #:028867 |
| *Ghsr* | Limited | *Le May et al., 2021* | |
| *Mc4r* | Extensive | This paper | JAX Strain #: 030759 |
| *Nts* | Extensive | This paper | JAX Strain #:017525 |
| *Ntsr1* | Extensive | This paper (previously unpublished; see Methods) | |
| *Oprm1* | Extensive | This paper, *Liu et al., 2022* | JAX Strain #:035574 |
| *Oxtr* | Moderate | *Ryan et al., 2017* | JAX Strain #: 030543 |
| *Pdyn* | Limited to extensive | This paper (previously unpublished; see Methods); *Grady et al., 2020*; *Huang et al., 2021a*; *Norris et al., 2021* | JAX Strain #: 927958 |
| *Penk* | Limited | *Norris et al., 2021* | JAX Strain #: 025112 |
| *Phox2b* | Extensive | This paper | JAX Strain #: 016223 |
| *Prlr* | Limited | *Kokay et al., 2018* | |
| *Ptger3* | Extensive | This paper | JAX Strain #:035575 |
| *Satb2* | Moderate to extensive | This paper; *Jarvie et al., 2021*; *Fu et al., 2019* | JAX Strain #: 030546 |
| Slc17a6 | Moderate to extensive | *Chiang et al., 2020*; *Grady et al., 2020*; *Huang et al., 2021b* | JAX Strain #: 028863 |
| *Slc32a1* | Extensive | This paper | JAX Strain #: 028862 |
| *Tac1* | Limited to extensive | This paper; *Barik et al., 2018* | JAX Strain #: 021877 |
| *Tacr1* | Limited to extensive | This paper (previously unpublished; see Methods); *Barik et al., 2021*; *Deng et al., 2020* | |

Cell populations in the PBN can be conceptually divided into two major groups based on overall projection patterns (*Figure 7A*). However, many of the lines did not neatly fit into one group, so the projections listed represent trends for the two pathways. The relatively dense projections from neurons in the PBle region generally follow the CTT, whereas the cells in the dorsal regions (PBld/ ls/lc) follow the VP as described by Geerling and colleagues (*Huang et al., 2021a*, *Huang et al., 2019*; *Karthik et al., 2022*). There is also a smaller periventricular pathway that passes through the periaqueductal gray (PAG, not shown) and a weak descending pathway to the hindbrain. None of the Cre-driver lines tested show projections exclusively along the periventricular or descending pathways. Some PBN neurons, e.g., *Tac1* neurons project along the descending pathway into the medulla

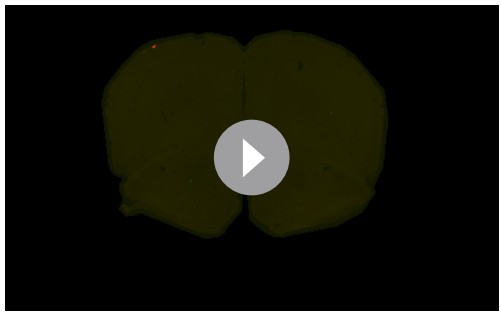

**Video 1.** Video showing the projection pattern of *Calca* neurons that reside in the PBN.
https://elifesciences.org/articles/81868/figures#video1

and affect escape-like behaviors and breathing (Arthurs et al. manuscript submitted; *Barik et al., 2018*). Qualitative scoring of synaptophysin expression in a selection of brain regions is shown (*Figure 7B*) based on relative fluorescence levels (strong, medium, weak, *Figure 7C*). Most neurons that reside in the PBmm are mixture cells that are also expressed in the lateral PBN; consequently, their axons join those from the lateral PBN.

For Cre-driver lines that have most of their expression in the PBle (*Calca, Crhr1, Nts, Chat*), or strongly in the PBle along with other regions (*Crh, Adcyap1, Adcyap1r1, Tac1, Mc4r, Oprm1*), the axon terminals with the brightest signal are in the BNST and CEA. Although expression is generally spread throughout these subregions, the oval and ventral regions of the BNST and the capsular region of the CEA have the strongest fluorescence. It is likely that the PBle cells are responsible for most of the oval/ventral BNST and capsular CEA staining density, while neurons outside of PBle project to adjacent regions. Although there is some variation, the next brightest regions are the IC, SI, VPMpc, and PSTN. There are no lines with cell bodies in the PBle that do not have at least some synaptophysin signal in the above listed regions. Other areas that are often innervated are the LS, CM, DMH, LHA, RE, IMD, PAG, DR, RN, and NTS. A complete list of abbreviations used here is shown in *Figure 4D*. Many of the Cre-driver lines that have most of their expression in PBle still have cells scattered throughout other subregions in the PBN, which may explain the expression in areas that have been associated with the ventral tract, such as the DMH and LHA.

There is overlap of axonal projections to both pathways, which likely occurs because none of the neuronal subclusters is restricted to one subregion of the PBN. Cre-driver lines that mainly have expression in the dorsal PBN regions (*Pdyn, Tacr1, Brs3, Cbln4, Ptger3*) also have axons that tend to travel through and target ventral brain regions such as the VTA, LHA, DMH, PVH, and MEPO. Cre-driver lines that are categorized into the dorsal group often have fewer cells and weaker projections as a result. Some Cre-driver lines have strong cellular expression across most of the lateral PBN. For these (*Adcyap1, Adcyap1r1, Oprm1, Crh*), there is robust expression of the AAV-driven fluorescent proteins in areas associated with the CTT such as the BNST/CEA and areas associated with the VP such as the MEPO. Overall, their projections are a combination of areas seen in the other groups.

The cells that reside in the PBmm do not appear to have a separate innervation profile. There was no line examined that had its expression limited to the PBmm exclusively. However, in one example (*Tac1*), the cells were transduced in PBmm on only one side, allowing for a comparison of projections between hemispheres. The result was a slightly brighter synaptophysin signal in the common projection targets of IC, BNST, and CEA, and a much brighter signal in the cortical amygdala on the side with the PBmm expression. In another example (*Phox2b*), most of the cells transduced were in the PBmm, with some cells in the PBlc and PBls. This line showed a projection to the septohippocampal (SH) nucleus, which likely originated from cells in PBls rather than PBmm because we did not see the same SH projection from the *Tac1* injection that was heavily expressed in the PBmm (compare *Phox2b* and *Tac1* files on Zenodo).

The *Satb2*-expressing neurons mainly reside in PBlv, scp, and PBmm. Although they are only partially expressed in PBle, their projection pathway still largely follows the CTT with connections to areas typically associated with PBle such as BNST and CEA. However, the axon terminals from *Calca*^Cre^-driver line are concentrated in the oval BNST and lateral CEA, whereas *Satb2*^Cre^-driven projections surround those regions (*Figure 8A–B*).

*Ntsr1* population of neurons has the VMH as its most prominent projection target (*Figure 8C*). A population of VMH-projecting PBN neurons was identified by *Fulwiler and Saper, 1984*; *Fulwiler and Saper, 1985*, who assigned it to the PBls subnucleus and found that about 80–90% of these neurons stained immunohistochemically for CCK. *Ntsr1* neurons are a subpopulation of the *Cck* cells in the far rostral PBN (*Figure 5—figure supplement 3*). The location of *Ntsr1* neurons is distinctly different

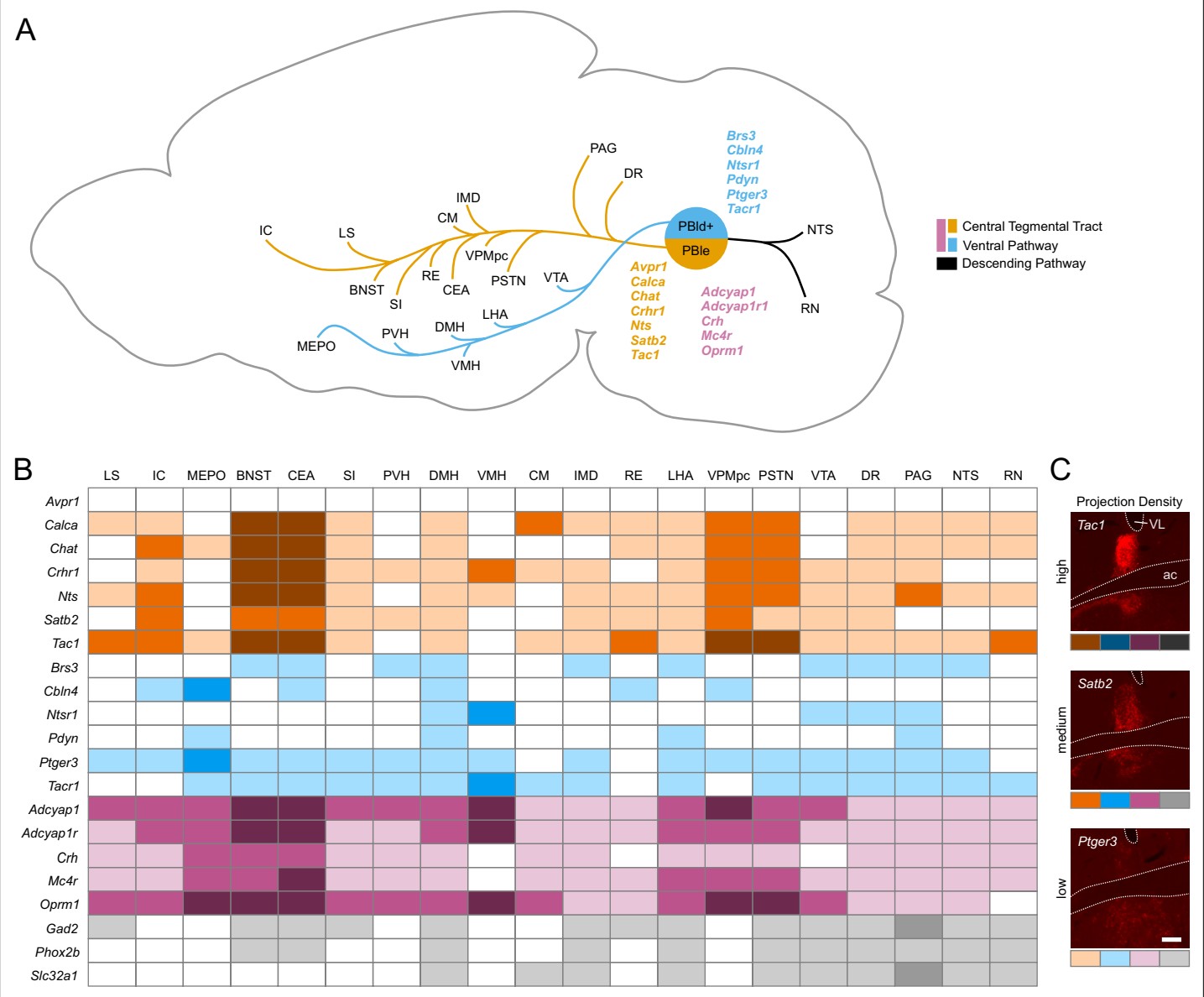

**Figure 7.** Descending pathways from the parabrachial nucleus (PBN) and surrounding regions and the strength of their projections. (**A**) Diagram showing the two main ascending projection pathways from the PBN, adapted from Figure 14 from *Huang et al., 2021b*. Genes are listed in a matched color with their pathway; pink genes follow both pathways. Many genes from each group have projections into the descending pathway that are not shown. (**B**) Guide showing approximate density of synaptophysin in a subset of target regions along with their abbreviations. Colors represent the pathways; darker shades indicate denser innervation.

from populations such as the *Tacr1* neurons in the PBls defined by the AMBA. All Cre-driver lines that have projections to the VMH (*Crhr1, Nts, Adcyap1, Adcyap1r1, Oprm1*) have some cell bodies in the same region as *Ntsr1*.

Many of the genes of interest are widely expressed in regions surrounding the PBN, which made restricting the transduction of cells within the PBN nearly impossible. *Gad2* and *Slc32a1*, the two GABAergic genes, are only sparsely expressed within the PBN compared to adjacent areas; consequently, we were unable to determine whether they function as interneurons or as projection neurons. *Gad2* is expressed more widely and without *Slc32a1* in some PBN glutamatergic cells, e.g., *Calca* neurons, so using that Cre-driver line to examine inhibitory projections outside of the PBN could be misleading.

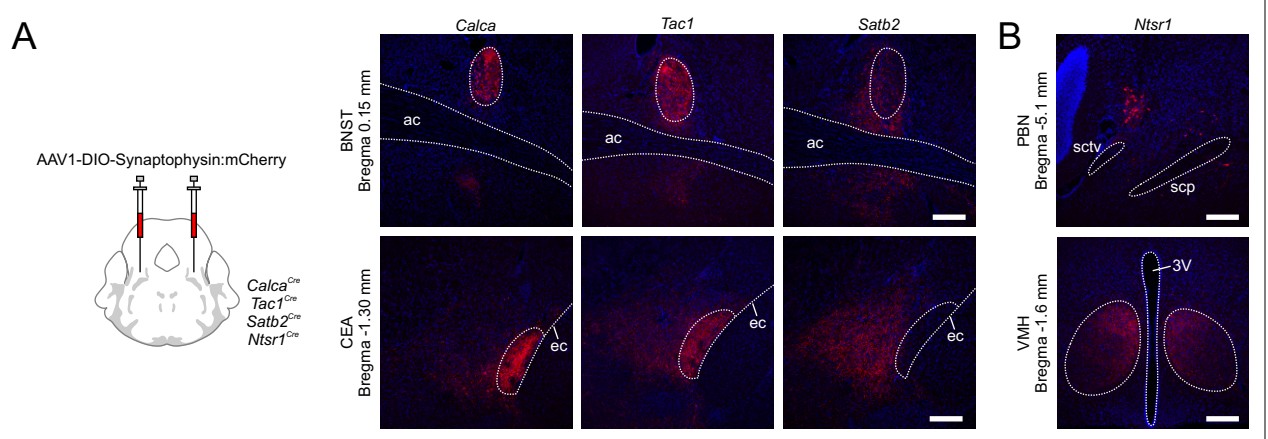

**Figure 8.** Projection patterns in target brain regions. (**A**) Comparison of synaptophysin:mCherry staining in bed nucleus of stria terminalis (BNST) and central nucleus of the amygdala (CEA) for *Calca, Tac1,* and *Satb2* Cre-driver lines. Scale bars, 200 µm. (**B**) Example of cell body location in nucleus of the lateral lemniscus region adjacent to parabrachial nucleus (PBN) for *Ntsr1^Cre^* mice (top); these neurons project almost exclusively to the ventromedial hypothalamus (VMH) (bottom). Scale bars, 200 µm.

## Discussion

The goals of this project were to establish a catalog of neuronal cell types within the PBN, characterize their transcriptional profiles, determine their location within the PBN, and map their axonal projections. The results should facilitate more rigorous delineation of how specific PBN populations respond to and relay interoceptive and exteroceptive signals. As expected, the round tissue punches used to isolate the PBN included some adjacent regions; consequently, some of the UMAP clusters are not in the PBN. We identified 13 glutamatergic subclusters within the PBN, whereas the GABAergic neurons are scattered throughout. Some of the excitatory neurons (e.g., *Calca, Pdyn, Tacr1*) were already known to primarily occupy distinct locations within the PBN and have distinct functions, but the present results reveal that expression is not restricted to specific PBN subregions. Most of the PBN neurons express multiple neuropeptides and GPCR receptors; the co-expressed transcription factors likely play an important role in their cell-specific gene expression. Many of the neuropeptides expressed in the PBN, e.g., cholecystokinin, substance P, CGRP, somatostatin, prodynorphin, neurotensin, were known to be expressed in the PBN based on immunohistochemistry studies and confirmed by in situ hybridization (*Hermanson et al., 1998*; *Saleh and Cechetto, 1996*; *Shimada et al., 1985*) The overall axonal projections of the PBN neurons were established using anterograde tracers and confirmed using S*lc17a6^Cre^* mice and injections of AAV carrying Cre-dependent fluorescent genes (*Huang et al., 2021a*). We provide detailed projection profiles from the PBN for many additional Cre-driver lines of mice that may generate ideas for testing their functions. The current data set provides a baseline for examining how the PBN changes during development and in response to environmental threats ranging from acute noxious events to chronic adverse conditions.

The PBN has been divided into 10 subregions based on rigorous Nissl cytoarchitectural criteria; subsequent anterograde and retrograde tracing studies revealed relatively distinct projections to and from different subregions of the PBN (see references in Introduction). As noted earlier, it is unfortunate that the nomenclature for different subregions of the PBN is not consistent in the literature, some regions are not shown at all in the AMBA, and a variety of abbreviations are used for the same subregions. Gene expression patterns provide an additional means to define these subregions as shown in *Figure 4*. Our results are consistent with those obtained by a new spatial-profiling method (PixelSeq) that was tested with PBN sections (*Fu et al., 2022*). We identified mRNAs with prominent expression in each of the PBN subregions except the PBme. Furthermore, what was once considered to be a unique population turns out to be more complex, e.g., both the *Calca* and *Pdyn* neurons are represented by two clusters based on scRNA-Seq analysis, and they could be subdivided even more by considering co-expression patterns of neuropeptides and other markers (*Huang et al., 2021a*, *Huang et al., 2021b*, *Karthik et al., 2022*). Whether these closely related clusters have distinct projections and functions needs to be established, e.g., are distinct *Pdyn* neurons involved in conveying

temperature, nocifensive behaviors, and feeding responses? The significance of the tight clustering of *Calca* neurons in rostral sections, with few if any interspersed neurons in the core of the PBle, is not known; it is unlikely that gap junctions allow simultaneous activation because the activities of individual neurons expressing a fluorescent calcium indicator (GCaMP) are not synchronized (*Chen et al., 2018*).

In contrast to the glutamatergic neurons, the GABAergic neurons are rare and dispersed throughout the PBN (*Raver et al., 2020*). GABAergic neurons in the lateral PBN and caudal KF project axons to the brainstem (*Geerling et al., 2017*; *Yokota et al., 2007*). Chemogenetic activation of GABAergic neurons in the PBN region can inhibit the function of local glutamatergic neurons (*Sun et al., 2020*). We have shown using electrophysiology that AAV-directed expression ChR2 in *Slc32a1^{Cre}* neurons in the PBN region can inhibit *Calca^{tdT}* neurons (Chen and Cao, unpublished).

There are several limitations to the conclusions reached in this study. The choice of parameters used for clustering of scRNA-Seq data can affect the number of clusters, and sequencing depth can influence the reliability of co-expression for low-abundance transcripts. For example, the number of cells in a neuronal subcluster expressing a particular GPCR mRNA (e.g., *Oprm1*) was low based on scRNA-Seq analysis, whereas the signal from the *Oprm1^{Cre}*-driver line revealed many cells, suggesting that this mRNA was not detected in most to the cells that were sequenced. We obtained ~100,000 reads/neuron which may not have been enough to detect rare transcripts. A single mRNA/cell can maintain ~10,000 proteins with a half-life of 700 min at steady state (see 'Estimating mRNA and protein abundance per cell' in Materials and Methods), which may be enough for many regulatory proteins. Deeper sequencing of more neurons and using multiple distinguishing probes may also reveal more cell types. Defining and distinguishing between different neuron types are challenging, requiring identification of master transcription factors along with multiple kinds of analysis including functional studies (*Zeng, 2022*). Thus, the results reported here will undoubtedly be refined by further investigation. Mice of both sexes were pooled for scRNA-Seq experiment, and only male mice were used for the HiPlex experiment, so future studies should consider this variable. Nevertheless, there was good, low-resolution correspondence among HiPlex, AMBA, and fluorescent protein expression from Cre-driver lines when transcripts were abundant, but we failed to detect a signal for some HiPlex probes (*Fn1, Mylk, Slc35d3, Shisal2b*) that were distinguishing genes based on scRNA-Seq analysis. There were also cases where predictions were not supported by HiPlex analysis (e.g., **N5, N6, N18**) even though scRNA-Seq indicated that these clusters express many of the same distinguishing transcription factors as other PBN clusters. Cre-driver lines are helpful for locating some neurons with low levels of gene expression (e.g., *Ntsr1, Avpr1a*) because, in principle, the action of a few Cre recombinase molecules is sufficient to activate robust expression from AAV carrying a Cre-dependent gene. When genes are expressed in the PBN as well as neighboring regions (e.g., *Slc32a1, Phox2b*), restricting AAV transduction to the PBN is challenging. It is also difficult to assure that any viral injection transduces all the relevant neurons within the PBN without doing an exhaustive analysis of each line. Animals and tissue sections were processed at different times, so variations in brightness could be due to aspects such as perfusion or staining quality rather than expression differences. The five coronal sections used for the HiPlex experiments did not include the most rostral region of the PBN, which explains our difficulty in locating clusters **N4** and **N9**. We did not include probes for some genes that are known to be expressed in the PBN and detected by scRNA-Seq (*Table 1*), e.g., *Penk* (*Engström et al., 2001*), *Grp* (*Karthik et al., 2022*), *Nmb* (*Huang et al., 2022*), and *Mc4r* (*Paues et al., 2006*). It will be important to determine whether any of them are co-localized with the in situ probes that we used.

The last decade has seen extensive use of Cre-driver lines of mice and AAV carrying Cre-dependent effector genes to interrogate the functions of PBN neurons. One strategy uses Cre-drivers with restricted expression. For example, numerous studies with *Calca^{Cre}*, *Pdyn^{Cre}*, and *Tacr1^{Cre}* mice have revealed their activation by a wide variety of real and potential threats, while their optogenetic or chemogenetic activation is generally aversive, and their inactivation ameliorates aversive responses to threats (references are included in Results, under **N10, N15, N17**). An alternative strategy uses Cre-driver mice with widespread expression in the PBN, e.g., *Slc17a6^{Cre}*, *Cck^{Cre}*, *Oprm1^{Cre}*, and *Tac1^{Cre}*, to assess behavioral and physiological consequences of their activation or inhibition (*Barik et al., 2018*; *Cheng et al., 2020*; *Chiang et al., 2020*; *Liu et al., 2022*; *Sun et al., 2020*). A problem with these Cre-drivers is the difficulty in restricting viral transduction

to the PBN. Both approaches have flaws because it is rare for threats to activate only one cell type, and all the cells represented by the more widely expressed genes are rarely engaged by specific threats. Furthermore, activation of some PBN neurons can suppress the activity of others, e.g., activation of *Tac1* neurons counteracts the outcomes of activating *Calca* neurons even though *Tac1* and *Calca* are co-expressed in the PBle (Arthurs et al. manuscript submitted). This possibility complicates interpretation of results when groups of neurons are artificially manipulated. One solution to this problem is tagging and manipulating groups of neurons that are normally engaged by specific threats, e.g., FosTrap (*DeNardo and Luo, 2017*; *Sakurai et al., 2016*) techniques have been used to identify PBN neurons activated by pain or aversive odors (*Liu et al., 2022*; *Rodriguez et al., 2017*).

The broad outlines of axonal projections from the PBN were established decades ago by anterograde tracing studies (*Fulwiler and Saper, 1984*; *Gauriau and Bernard, 2002*; *Krout and Loewy, 2000*; *Moga et al., 1990*; *Norgren and Leonard, 1971*; *Tokita et al., 2009*) and confirmed by injection of AAV-expressing Cre-dependent fluorescent markers in the PBN of *Slc17a6^Cre* mice (*Chiang et al., 2020*; *Huang et al., 2021a*). The axonal tracts to the forebrain follow two main paths as they leave the PBN. The CTT pathway travels to the forebrain via the ventral thalamus, to the extended amygdala and cerebral cortex. The VP travels through the ventral tegmental area to the hypothalamus. A periventricular pathway passing through the PAG and innervates regions such as the PVT also exists (*Huang et al., 2021b*) but was not explored here because no line projected exclusively through it, and it was difficult to separate hypothalamic projections from those originating from the VP. A descending pathway innervates parts of the hindbrain including the NTS, pre-Bötzinger, and reticular regions. *Calca* neurons have axon collaterals that go to more than one brain region (*Bowen et al., 2020*). Most of the Cre-driver lines tested here resemble either the *Calca* neurons or the *Pdyn* neurons in their axonal projections; however, there are distinct differences in their innervation of forebrain targets that is revealed by synaptophysin. We identified *Ntsr1* neurons as a restricted population in a distinct region of far rostral PBN with axons that travel along the VP to innervate mainly the VMH. This makes *Ntsr1^Cre* mice potentially useful for studying that connection in more detail.

The axonal inputs to specific clusters of PBN neurons are being established, either by retrograde rabies virus tracing studies starting with specific Cre-drivers, e.g., *Calca^Cre* (*Liu et al., 2022*; *Rodriguez et al., 2017*) or by candidate approaches starting with expression of AAV-DIO-ChR2 into distal sites of Cre-driver mouse lines mice and recording photoactivated currents in specific PBN neurons, e.g., input from *Cck* or *Dbh* neurons in the NTS to *Calca* neurons in the PBN (*Roman et al., 2016*), *Oxt* neurons in preoptic area to *Oxtr* neurons in PBN (*Ryan et al., 2017*), *Slc17a6* or *Slc32a1* inputs from BNST to *Pdyn* and *Calca* neurons in PBN (*Luskin et al., 2021*). Many other molecularly defined inputs to different subregions of the lateral PBN have been described, e.g., *Tac1*, *Tacr1*, and *Gpr83* from spinal cord (*Choi et al., 2020*), *Gfral* and *Glp1r* from area postrema (*Zhang et al., 2021*), *Calcr* and *Tac1* from NTS (*Cheng et al., 2020*; *Xie et al., 2022*), *Slc17a6* and *Dbh* from LC (*Yang et al., 2021*), *Slc6a3* from ventral tegmental area (*Han et al., 2021*), *Slc32a1* from substantia nigra reticulata, *Npy* and *Slc32a1* from arcuate nucleus (*Alhadeff et al., 2018*; *Wu et al., 2009*); *Mc4r* from the PVN (*Garfield et al., 2015*), *Htr2a*, *Prkcd*, or *Sst*, *Pdyn*, and *Crh* from CEA (*Cai et al., 2014*; *Douglass et al., 2017*; *Raver et al., 2020*). In the latter cases, knowing the locations and molecular identity of PBN clusters that they innervate would refine connectivity maps and provide insight into potential functions.

Our study highlights the diversity of cell types in the PBN, many of which reside in distinct subregions that align with prior anatomical tracing studies. We also mapped the PBN cell distribution and detailed brain-wide projections of 21 Cre-driver mouse lines. All these data are publicly available for download, and most of the mice have already been deposited at the Jackson Laboratory. This rich resource will inspire and inform future studies on the role and neurocircuitry of the PBN.

## Materials and methods

**Key resources table**

| Reagent type (species) or resource | Designation | Source or reference | Identifiers | Additional information |
|---|---|---|---|---|
| Strain, strain background (*Mus musculus*) | Refer to *Table 3* for sources of all Cre-driver lines of mice. | | | |
| Strain, strain background (*AAV1*) | pAAV1-Ef1α-DIO-YFP | Karl Deisseroth | Addgene Plasmid #27056 RRID:Addgene_27056 | |
| Strain, strain background (*AAV1*) | pAAV1-Ef1α-DIO-Synaptophysin-mCherry | *Roman et al., 2016* | | |
| Antibody | Anti-GFP (chicken polyclonal) | Abcam | Cat#: ab13970 RRID:AB_300798 | 1:10,000 |
| Antibody | Anti-dsRed (rabbit monoclonal) | Takara | Cat#: 632496 RRID:AB_10013483 | 1:1,000 |
| Antibody | Alexa Fluor 488 anti-chicken (donkey monoclonal) | Jackson ImmunoResearch | Cat#: 703-545-155 RRID:AB_2340375 | 1:500 |
| Antibody | Alexa Fluor 494 anti-rabbit (donkey monoclonal) | Jackson ImmunoResearch | Cat#: 711-585-152 RRID:AB_2340621 | 1:500 |
| Commercial assay or kit | RNAscope HiPlex12 Reagents Kit (488, 550, 647) | Advanced Cell Diagnostics | Cat No. 324108 | |
| Commercial assay or kit | RNAscope HiPlex Probe- Mm-Brs3-T2 | Advanced Cell Diagnostics | Cat No. 454111-T2 | |
| Commercial assay or kit | RNAscope HiPlex Probe- Mm-Calcr-T6 | Advanced Cell Diagnostics | Cat No. 494071-T6 | |
| Commercial assay or kit | RNAscope HiPlex Probe- Mm-Crh-T9 | Advanced Cell Diagnostics | Cat No. 316091-T9 | |
| Commercial assay or kit | RNAscope HiPlex Probe- Mm-Fn1-T4 | Advanced Cell Diagnostics | Cat No. 316951-T4 | |
| Commercial assay or kit | RNAscope HiPlex Probe- Mm-Gal-T8 | Advanced Cell Diagnostics | Cat No. 400961-T8 | |
| Commercial assay or kit | RNAscope HiPlex Probe- Mm-Ghrh-T3 | Advanced Cell Diagnostics | Cat No. 470991-T3 | |
| Commercial assay or kit | RNAscope HiPlex Probe- Mm-Nfib-T4 | Advanced Cell Diagnostics | Cat No. 586511-T4 | |
| Commercial assay or kit | RNAscope HiPlex Probe- Mm-Nmu-T5 | Advanced Cell Diagnostics | Cat No. 446831-T5 | |
| Commercial assay or kit | RNAscope HiPlex Probe- Mm-Npnt-T11 | Advanced Cell Diagnostics | Cat No. 316771-T11 | |
| Commercial assay or kit | RNAscope HiPlex Probe- Mm-Pappa-T5 | Advanced Cell Diagnostics | Cat No. 443921-T5 | |
| Commercial assay or kit | RNAscope HiPlex Probe- Mm-Pax5-T6 | Advanced Cell Diagnostics | Cat No. 541761-T6 | |
| Commercial assay or kit | RNAscope HiPlex Probe- Mm-Pdyn-T7 | Advanced Cell Diagnostics | Cat No. 318771-T7 | |
| Commercial assay or kit | RNAscope HiPlex Probe- Mm-Piezo2-O1-T12 | Advanced Cell Diagnostics | Cat No. 500501-T12 | |
| Commercial assay or kit | RNAscope HiPlex Probe- Mm-Pla2g2f-O1-T1 | Advanced Cell Diagnostics | Cat No. 1006331-T1 | |
| Commercial assay or kit | RNAscope HiPlex Probe- Mm-Pnoc-T2 | Advanced Cell Diagnostics | Cat No. 437881-T2 | |
| Commercial assay or kit | RNAscope HiPlex Probe- Mm-Satb2-T10 | Advanced Cell Diagnostics | Cat No. 413261-T10 | |

| Reagent type (species) or resource | Designation | Source or reference | Identifiers | Additional information |
|---|---|---|---|---|
| Commercial assay or kit | RNAscope HiPlex Probe- Mm-Slc17a7-T3 | Advanced Cell Diagnostics | Cat No. 416631-T3 | |
| Commercial assay or kit | RNAscope HiPlex Probe- Mm-Slc32a1-T1 | Advanced Cell Diagnostics | Cat No. 319191-T1 | |
| Commercial assay or kit | RNAscope HiPlex Probe- Mm-Slc32a1-T8 | Advanced Cell Diagnostics | Cat No. 319191-T8 | |
| Commercial assay or kit | RNAscope HiPlex Probe- Mm-Sostdc1-T12 | Advanced Cell Diagnostics | Cat No. 313151-T12 | |
| Commercial assay or kit | RNAscope HiPlex Probe- Mm-Stk32b-T11 | Advanced Cell Diagnostics | Cat No. 564841-T11 | |
| Commercial assay or kit | RNAscope HiPlex Probe- Mm-Th-T9 | Advanced Cell Diagnostics | Cat No. 317621-T9 | |
| Commercial assay or kit | RNAscope HiPlexUp Reagent | Advanced Cell Diagnostics | Cat No. 324190 | |
| Commercial assay or kit | RNAscope Probe - Mm-Calca-alltv-C2 | Advanced Cell Diagnostics | Cat No. 417961-C2 | |
| Commercial assay or kit | RNAscope Probe - Mm-Cck-C3 | Advanced Cell Diagnostics | Cat No. 402271-C3 | |
| Commercial assay or kit | RNAscope Probe - Mm-Gal-C2 | Advanced Cell Diagnostics | Cat No. 400961-C2 | |
| Commercial assay or kit | RNAscope Probe - Mm-Gda-C3 | Advanced Cell Diagnostics | Cat No. 520531-C3 | |
| Commercial assay or kit | RNAscope Probe - Mm-Nps | Advanced Cell Diagnostics | Cat No. 485201 | |
| Commercial assay or kit | RNAscope Probe – Mm-Ntsr1-C2 | Advanced Cell Diagnostics | Cat No. 422411-C2 | |
| Commercial assay or kit | RNAscope Probe- Mm-Pdyn-C3 | Advanced Cell Diagnostics | Cat No. 318771-C3 | |
| Commercial assay or kit | RNAscope Probe- Mm-Satb2 | Advanced Cell Diagnostics | Cat No. 413261 | |
| Commercial assay or kit | RNAscope Probe - Mm-Slc17a6 | Advanced Cell Diagnostics | Cat No. 319171 | |
| Commercial assay or kit | RNAscope Probe- Mm-Tac1-C3 | Advanced Cell Diagnostics | Cat No. 410351-C3 | |
| Commercial assay or kit | RNAscope Probe- Mm-Th-C3 | Advanced Cell Diagnostics | Cat No. 317621-C3 | |
| Commercial assay or kit | RNAscope Fluorescent Multiplex Reagent Kit | Advanced Cell Diagnostics | Cat No. 320850 | |
| Commercial assay or kit | Chromium Single Cell Controller & Accessory Kit | 10 X Genomics | Cat # 120263 | |
| Commercial assay or kit | Chromium Single Cell 3' Library and Gel Bead Kit v2 | 10 X Genomics | Cat # 120267 | |
| Commercial assay or kit | Chromium Single Cell A Chip Kit | 10 X Genomics | Cat # 120236 | |
| Commercial assay or kit | Chromium i7 Multiplex Kit | 10 X Genomics | Cat # 120262 | |
| Commercial assay or kit | Dead Cell Removal Kit | Miltenyi Biotec | Cat # 130-090-101 | |
| Commercial assay or kit | Illumina HiSeq | Genewiz | | |
| Commercial assay or kit | SPRIselect | Beckman Coulter | Product No: B23317 | |
| Commercial assay or kit | 4200 TapeStation | Agilent | G2991AA | |
| Commercial assay or kit | High Sensitivity D5000 ScreenTape | Agilent | Part Number: 5067–5592 | |
| Other | Normal donkey serum | Jackson ImmunoResearch | Cat#:017-000-121 RRID:AB_2337258 | See: Materials and Methods Stereotaxic Surgery and Projection Tracing |
| Software, algorithm | RNAscope HiPlex Image Registration Software | Advanced Cell Diagnostics | Cat # 300065 | |

| Reagent type (species) or resource | Designation | Source or reference | Identifiers | Additional information |
|---|---|---|---|---|
| Software, algorithm | FIJI | ImageJ | RRID: SCR_002285 | |
| Software, algorithm | NeuroInfo | MBF Bioscience | | |
| Software, algorithm | R | https://www.r-project.org/; RRID: SCR_001905 | | |
| Software, algorithm | Seurat v3.1.2 | https://Github.com/JonathanShor/DoubletDetection; RRID:SCR_016341 | | |
| Software, algorithm | bcl2fastq v2.18.0.12 | https://www.illumina.com/; RRID: SCR_015058 | | |
| Software, algorithm | Cell Ranger v3.1.0 | 10 X Genomics; RRID: SCR_017344 | | |
| Software, algorithm | SAMtools v1.10 | https://www.htslib.org/; RRID: SCR_002105 | | |
| Software, algorithm | Python | https://www.python.org/; RRID: SCR_008394 | | |

## Mice

All experiments were approved by the Institutional Animals Care and Use Committee at the University of Washington (Protocol #2183–02). Animals were group-housed with littermates on a 12-hr light cycle at ~22°C with food and water available ad libitum. Male and female mice from the same litter were used, but no comparisons were done between sex. Refer to *Table 3* for sources of all Cre-driver lines of mice. Most of them have been described previously and/or deposited at Jackson labs, where details can be found. All mice were on a C57BL/6J genetic background. We have not documented that each of the Cre-driver lines tested is restricted to neurons that normally express that gene in the adult.

The generation of *Calca^tdTomato* mice was described (*Jarvie et al., 2021*). The generation of four new lines is described below. The *Cbln4, Ntsr1,* and *Tacr1* Cre-driver lines were made by inserting IRES-Cre:GFP just beyond the termination codon of these genes, whereas the *Pdyn* line was made by inserting Cre:GFP just 5' of the initiation codon. The 5' and 3' arms (5–8 kb) were amplified from C57BL/6 BAC clones by PCR using Q5 polymerase (New England Biolabs) and inserted into a cloning vector that contains frt-flanked SV-Neo for positive selection and *Pgk*-DTA and HSV-TK genes for negative selection (*Jarvie et al., 2021*). The linearized constructs were electroporated in G4 ES cells (129× C57B/L6). About 80 G418-resistant clones were picked and expanded for Southern blot analysis using a $^{32}$P-labeled probe located just beyond either the 5' or 3' arm. Single inserts were established by southern blot with a Neo gene probe. Correctly targeted clones were injected into blastocysts and transferred to recipient female mice. Germline transmission of the targeted allele was determined by three-primer PCR (two primers from gene on interest that flanking Cre insertion region and one reverse primer in IRES or Cre). Mice were bred with *Gt(Rosa26)-FLPo* (Jax: 07844) mice to remove the SV-Neo gene and then bred with C57BL/6 mice for at least six generations.

## Single-cell library preparation and sequencing

Live, single-cell suspensions were prepared as described (*Rossi et al., 2021*; *Rossi et al., 2019*), and tissue was harvested approximately 6 hr after the onset of the dark cycle (Zeitgeber time ~18:00). Briefly, male and female mice were transcardially perfused with a cold artificial CSF solution containing N-methyl-D-glucamine (NMDG) (NMDG-aCSF), modified from *Ting et al., 2018*. All steps were performed under continuous oxygenation and $CO_2$ buffering using 95%/5% $O_2$/$CO_2$. Brains were rapidly dissected, and coronal slices (200 μm) spanning the PBN were prepared using a vibrating microtome (VT1200, Leica Biosystems). Slices were allowed to recover in NMDG-aCSF containing 500-nM TTX, 10-μM AP-V, and 10-μM DNQX (NMDG-aCSF-R) for 20 min, and the PBN was subsequently isolated using tissue punches (500–750 μm). The isolated tissue was enzymatically digested using 1 mg/mL pronase (Roche) for 50 min, and enzymatic digestion was quenched with 0.05% BSA. Cells were mechanically dissociated using a fire-polished glass pipet with an internal diameter of 200–300 μm, filtered through a 40 μm strainer, washed, and depleted of dead cellular fragments

using a commercial kit (Miltenyi Biotec, Bergisch Gladbach, Germany). The remaining cells were resuspended in PBS containing 0.05% BSA about 1000 cells/μL.

Single-cell RNA libraries were generated using chromium single-cell 3' v2 chemistry (10× Genomics, Pleasanton, CA) following the standard manufacturer protocol. A pool of ~17,000 cells harvested from 5 mice were loaded per reaction, with the first pool run on a single reaction and the second pool spread across three reactions (*Figure 1—figure supplement 1H–I*).

For each library, cDNA amplification was performed using 12 cycles, and indexing was performed using 11 cycles. Library size distribution and concentration were determined using High Sensitivity D5000 ScreenTape or 4200 TapeStation (Agilent Technologies, Santa Clara, CA) and Qubit HS DNA Assay (Invitrogen, Waltham, MA). Each reaction library was sequenced on two lanes of an Illumina HiSeq 4000 using 2×150 chemistry by Genewiz, Inc (South Plainfield, NJ) following the standard 10× Genomics v2 paired-end configuration. Approximately 875 million reads were generated per lane with a mean sequencing saturation of 82.6%. Sequences were aligned to the mm10-3.0.0 genome, and digital expression matrices were created using 10× Genomics Cell Ranger v3.1.0 with 128 GB of memory on 24 cores.

## Single-cell clustering analysis and feature discovery

Clustering was performed using Seurat v3.1.2 (*Stuart et al., 2019*) and custom code in R v3.6.1 as described (*Rossi et al., 2021*). Briefly, to remove low-quality cells, total input cells were first filtered using a threshold for genes and fraction mitochondrial reads, and doublets were subsequently removed using DoubletDetection v2.5.4 under default parameters on Python 3.7 via Reticulate v1.14 (*Gayoso and Shor, 2022*). Cells containing ≤800 genes and ≥10% mitochondrial reads were removed from the primary analysis (*Figure 1* and *Figure 1—figure supplement 1*). Within the analysis of all cells, one subcluster was unable to be mapped to any specific features and was excluded from analysis (*Figure 1B*, gray). Following these filters, a total of 39,649 cells were retained with a median of 1740 genes and 3366 transcripts represented across a median of 47,177 reads per cell (*Figure 1* and *Figure 1—figure supplement 1A–E*). The neuronal cluster identified in the initial analysis of all cells (*Figure 1B*) was then isolated and subjected to a more stringent quality threshold where only cells containing ≤2% mitochondrial reads were retained to enable high-confidence, high-resolution subclustering (*Figure 2—figure supplement 1C and H*). Within the neurons, two subclusters were unable to be mapped to any specific features and were excluded from the analysis (*Figure 2A*, gray). This resulted in a final analysis of 7635 neurons containing a median of 3189 genes, 7823 transcripts, and 99,583 reads per cell (*Figure 2* and *Figure 2—figure supplement 1A–J*).

Regression and integration of samples were performed using a regularized negative binomial regression (*Hafemeister and Satija, 2019*; *Stuart et al., 2019*) and canonical correlation analysis (*Butler et al., 2018*; *Stuart et al., 2019*) as described (*Rossi et al., 2021*; *Figure 1—figure supplement 1F–G* and *Figure 2—figure supplement 1K–M*). Principal components were calculated on all genes, and cells were plotted in UMAP space using iteratively tuned parameters to optimize for visualization (*Mcinnes et al., 2018*; *Figure 1—figure supplement 1J* and *Figure 2—figure supplement 1O*). Clustering was performed using the Louvain algorithm with multilevel refinement (*Rodriguez and Laio, 2014*) with a K parameter of 25 and resolution of 0.04 (all cells) (*Figure 1*), or a K parameter of 15 and resolution of 0.25 (neurons) (*Figure 2*). Feature discovery for cell-type assignment was performed on Pearson residuals using a likelihood ratio test for single-cell data as implemented in Seurat (*Hafemeister and Satija, 2019*; *Macosko et al., 2015*; *McDavid et al., 2013*).

Prepossessing, regression, integration, dimensionality reduction, clustering, and feature discovery were run on a Dell blade-based cluster at the University of North Carolina at Chapel Hill running Linux RedHat Enterprise 7.7. All other steps were performed on an Apple MacBook Pro running macOS 11.4.0. Raw and processed data for the scRNA-seq experiment have been deposited at the National Center for Biotechnology Information Gene Expression Omnibus (NCBI GEO, accession number GSE207708). The code used in this analysis is available at a Github repository affiliated with the Stuber Laboratory group (https://github.com/stuberlab/Pauli-Chen-Basiri-et-al-2022; *Basiri, 2022*).

## RiboTag analysis of transcripts enriched in *Calca* neurons

The *Calca^Cre* mice were used to identify mRNAs that were selectively being translated in neurons that express the *Calca* gene by injecting a virus expressing a Cre-dependent hemagglutinin (HA)-tagged

ribosomal protein 22 (AAV-DIO-Rpl22-HA) (*Sanz et al., 2015*) into the PBN of eight adult mice. After several weeks to allow incorporation of the tagged ribosomal protein into ribosomes, tissue punches (two pools of bilateral punches from four mice) were collected, total cell extract was prepared, and then polyribosomes were precipitated with an antibody against the HA-tagged ribosomes (*Sanz et al., 2019*; *Sanz et al., 2009*). For microarray analysis, 10 ng of total RNA was amplified and biotin-labeled using the Ovation Pico SL WTA system with the EncoreIL biotinylation module (NuGEN), and 750 ng of the labeled cDNA was hybridized to a MouseRef-8v2.0 gene expression BeadChip (Illumina). Signals were detected using the BeadArray Reader (Illumina) and analyzed using the GenomeStudio software (Illumina). Average normalization and the Illumina custom error model were applied to the analysis. Only transcripts with a differential score of >13 ($p<0.05$) were considered. The results shown are means of two biological replicates. Raw and normalized RiboTag data have been deposited in the NCBI GEO (accession number GSE207153). In some cases, the microarray included more than one probe for the same gene, and the absolute values could differ greatly; thus, the relative enrichment is more reliable.

## Stereotaxic surgery and projection tracing

Mice were anesthetized with isoflurane and placed on a robotic stereotaxic frame (Neurostar GmbH, Tübingen, Germany). AAV1-EF1a-DIO-YFP and AAV1-EF1a-DIO-synaptophysin:mCherry were injected bilaterally into the PBN (AP –4.8 mm, ML ±1.4 mm, DV 3.5 mm) at a rate of 0.1 µl/min for 2 min. At least 3 weeks after virus injection, mice were deeply anesthetized with sodium pentobarbital and phenytoin sodium (0.2 ml, i.p.) and intracardially perfused with ice-cold PBS followed by 4% PFA. Brains were post-fixed overnight in 4% PFA at 4°C, cryoprotected in 30% sucrose, frozen in OCT compound, and stored at –80°C. Coronal sections (35 µm) spanning the brain (Bregma 2.0 to –8.0 mm) were cut on a cryostat and collected in cryoprotectant for long-term storage at –20°C. The YFP and mCherry signals overlapped virtually in all neurons, indicative of co-infection by both AAV. The synaptophysin mCherry signal helps to distinguish axon terminals from fibers of passage. The two colors can be visualized separately or together in the TIFF stacks on Zenodo. Although AAV1 serotype has been shown to cross synapses in an anterograde direction, we never observed fluorescence in post-synaptic cells.

Sections were washed two times in PBS and incubated in a blocking solution (3% normal donkey serum and 0.2% Triton X-100 in PBS) for 1 hr at room temperature. Sections were incubated overnight at 4°C in blocking solution with primary antibodies including: chicken-anti-GFP (1:10,000) and rabbit-anti-dsRed (1:2000). After three washes in PBS, sections were incubated for 1 hr in PBS with secondary antibodies: Alexa Fluor 488 donkey anti-chicken and Alexa Fluor 594 donkey anti-rabbit, (1:500). Tissue was washed three times in PBS, mounted onto glass slides, and coverslipped with Fluoromount-G with DAPI (Southern Biotech).

Whole-slide fluorescent images were acquired using a Keyence BZ-X710 microscope and higher magnification images using an Olympus FV-1200 confocal microscope. Images were minimally processed using Fiji to enhance brightness and contrast for optimal representation of the data. For TIFF stacks, images were aligned using the BrainMaker workflow in NeuroInfo (MBF Bioscience).

## RNAscope multiplex/HiPlex FISH

Male mice were deeply anesthetized with sodium pentobarbital and phenytoin sodium (0.2 ml, i.p.), decapitated, and brains rapidly frozen on crushed dry ice. Coronal sections (15 µm) were cut on a cryostat, mounted onto SuperFrost Plus slides, and stored at –80°C. RNAscope HiPlex assay or RNAscope fluorescent multiplex assay was performed following the manufacturer's protocols. For one experiment, two AAV-SaCas9 with guide RNAs directed against different parts of *Slc17a6* (*Hunker et al., 2020*) were injected into the PBN 16 weeks prior to RNAScope for *Calca* and *Slc17a6*.

Images centered on the scp in the PBN were acquired in a 3×3 grid at 20× using a Keyence BZ-X710 microscope then stitched together using Fiji. Images of probe staining within the four-channel sets were subtracted from one another using Fiji's image calculator function to remove background autofluorescence. The DAPI images from each of the four sets of images were registered using the HiPlex Image Registration Software (ACDBio) and then used to align all the probe images. This process was repeated for a second HiPlex experiment. Colors were assigned using the HiPlex Image Registration Software.

## Evaluation of gene expression and projection density

Registered HiPlex probe images were combined into five stacks for each Bregma level for both experiments. PBN and surrounding subregions of interest (ROIs) were drawn based on the AMBA designations and distinct probe locations. We recognize that the boundaries of the PBN in the AMBA may not accurately reflect the cytoarchitecture. It was used as a reference to allow readers to locate mRNA expression more easily. Using these ROIs, we generated a score based on an estimation of the number of transcripts present per cell and number of cells labeled per region. Projection regions were evaluated based on synaptophysin density by matching the brain sections in the TIFF stacks as closely to the AMBA as possible. The values shown in the tables are only meant to be a guide, and all the raw data are available for interested readers to evaluate on their own.

## Estimating mRNA and protein abundance per cell

Using an RNA/DNA ratio of 2 for total brain (probably an underestimate for neurons), a DNA content of 6.4 pg/cell and about 80% of total RNA being ribosomal RNA, there is about 10 pg of ribosomal RNA per brain cell. Ten pg rRNA/cell corresponds to about 1 million ribosomes/cell. With 10 ribosomes per mRNA that equals 100,000 mRNAs/cell. At steady state, the amount of protein synthesized per mRNA equals rate of synthesis times the half-life divided by the natural log of 2 (0.67). The rate of synthesis equals the amount of mRNA times the translational efficiency (number of proteins made/min/mRNA). With a translational efficiency of 10, and a half-life of 700 min, one mRNA per cell can maintain about 10,000 proteins/cell at steady state.

## Acknowledgements

We thank Susan Phelps for maintaining the mouse lines used in these studies, and Dr. Avery Hunker and Dr. Larry Zweifel for providing the AAV-SaCas9 with guide RNAs against *Slc17a6*. We appreciate the comments from lab members and reviewers. We also thank Dr. Clif Saper and especially Dr. Joel Geerling for his extensive conceptual and editorial suggestions. This work was supported in part by grants from the National Institutes of Health, R01-DA24908 (RDP), R01-DA032750, and R01-DA038168 (GDS).

## Additional information

### Competing interests

Richard D Palmiter: Reviewing editor, *eLife*. The other authors declare that no competing interests exist.

### Funding

| Funder | Grant reference number | Author |
|---|---|---|
| National Institutes of Health | R01-DA24908 | Richard D Palmiter |
| National Institutes of Health | R01-DA032750 | Garret D Stuber |
| National Institutes of Health | R01-DA038168 | Garret D Stuber |

The funders had no role in study design, data collection and interpretation, or the decision to submit the work for publication.

### Author contributions

Jordan L Pauli, Data curation, Investigation, Visualization, Writing - review and editing; Jane Y Chen, Data curation, Investigation, Visualization, Project administration, Writing - review and editing; Marcus L Basiri, Data curation, Formal analysis, Investigation, Visualization, Writing - review and editing; Sekun Park, Data curation, Investigation, Writing - review and editing; Matthew E Carter, Investigation; Elisenda Sanz, Investigation, Writing - review and editing; G Stanley McKnight, Supervision, Funding

acquisition; Garret D Stuber, Supervision, Funding acquisition, Writing - review and editing; Richard D Palmiter, Conceptualization, Supervision, Funding acquisition, Writing - original draft, Project administration, Writing - review and editing

## Author ORCIDs
Jordan L Pauli http://orcid.org/0000-0001-6276-3407
Jane Y Chen http://orcid.org/0000-0002-3986-8785
Marcus L Basiri http://orcid.org/0000-0002-4829-7187
Matthew E Carter http://orcid.org/0000-0003-1802-090X
Elisenda Sanz http://orcid.org/0000-0002-7932-8556
Garret D Stuber http://orcid.org/0000-0003-1730-4855
Richard D Palmiter http://orcid.org/0000-0001-6587-0582

## Ethics
All experiments were approved by the Institutional Animals Care and Use Committee at the University of Washington (Protocol #2183-02).

## Decision letter and Author response
Decision letter https://doi.org/10.7554/eLife.81868.sa1
Author response https://doi.org/10.7554/eLife.81868.sa2

---

# Additional files

## Supplementary files
• Supplementary file 1. Table denoting the average normalized expression, fraction of cells expressing, and likelihood ratio p-value for every gene in each cluster.

• Supplementary file 2. Table denoting the average normalized expression, fraction of cells expressing, and likelihood ratio p-value for every gene in each neuronal subcluster.

• Supplementary file 3. Expression of neuropeptides across neuronal subclusters plotted according to their average normalized expression and fraction of cells expressing each gene.

• Supplementary file 4. Expression of G-protein-coupled receptors across neuronal subclusters plotted according to their average normalized expression and fraction of cells expressing each gene.

• Supplementary file 5. HiPlex data for all probes at five Bregma levels of the parabrachial nucleus.

• Supplementary file 6. Data for Ribotag experiment showing all genes (1) and genes significantly enriched/depleted (p<0.05) sorted by fold change (FC, 2).

• MDAR checklist

## Data availability
Raw and preprocessed data for scRNA-seq: NCBI GEO accession number GSE207708. Code for analysis of scRNA-Seq data: https://github.com/stuberlab/Pauli-Chen-Basiri-et-al-2022, (copy archived at swh:1:rev:8e974c4655cc4e4f1f3ce853b793af86f0e876bf). Raw and normalized data for RiboTag: NCBI GEO accession number GSE207153. Images from RNAscope and all tracing experiments: Zenodo DOI: https://doi.org/10.5281/zenodo.6707404.

The following datasets were generated:

| Author(s) | Year | Dataset title | Dataset URL | Database and Identifier |
|---|---|---|---|---|
| Chen JY, Basiri ML, Stuber GD, Palmiter RD | 2022 | Molecular and Anatomical Characterization of Parabrachial Neurons and Their Axonal Projections | https://www.ncbi.nlm.nih.gov/geo/query/acc.cgi?acc=GSE207708 | NCBI Gene Expression Omnibus, GSE207708 |

*Continued on next page*

*Continued*

| Author(s) | Year | Dataset title | Dataset URL | Database and Identifier |
|---|---|---|---|---|
| Sanz E, McKnight GS | 2022 | Gene expression profiling of Calca neurons in the parabrachial nucleus (PBN) | https://www.ncbi.nlm.nih.gov/geo/query/acc.cgi?acc=GSE207153 | NCBI Gene Expression Omnibus, GSE207153 |
| Pauli JL, Chen JY, Palmiter RD | 2022 | Molecular and Anatomical Characterization of Parabrachial Neurons and Their Axonal Projections | https://doi.org/10.5281/zenodo.6707404 | Zenodo, 10.5281/zenodo.6707404 |

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
