## [Editor Report]

The parabrachial nuclei are groups of neurons in the brainstem (one on each side) that integrate information about the state of the body to guide appropriate behavioral and homeostatic responses. The manuscript by Pauli and Chen et al. is a beautiful and much-needed study that characterizes the cell types that make up these nuclei. The result is a highly valuable resource to the academic community.

---

## [Decision Letter]

**Decision letter after peer review:**

Thank you for submitting your article "Molecular and Anatomical Characterization of Parabrachial Neurons and Their Axonal Projections" for consideration by *eLife*. Your article has been reviewed by 3 peer reviewers, and the evaluation has been overseen by a Reviewing Editor and Catherine Dulac as the Senior Editor. The following individuals involved in review of your submission have agreed to reveal their identity: Asaf Keller (Reviewer #1); Seungwon Choi (Reviewer #2); Sarah E Ross (Reviewer #3).

The reviewers have discussed their reviews with one another, and the Reviewing Editor has drafted this to help you prepare a revised submission. Although no new experiments are required for publication, a revised paper should address the following points.

Essential revisions:

1. Although both males and females were used, the authors state that "no formal comparisons were done between sex". Many of the peptides and receptors identified here exhibit sexual dimorphism. Information on potential sex differences would be valuable, and the authors should comment on this.

2. Geerling and collaborators (PMC9119955) have recently published a catalog of cell populations in PB, focusing primarily, but not exclusively, on Atoh1 and Lmx1. The present authors have cited this paper more than once in this manuscript. However, it might be useful for the readers to relate the conclusions of the present manuscript to those presented by Geerling's group.

3. In the abstract, the authors describe the PBN as being involved in pain sensation. Many people in the pain field would cringe at this description since they feel that pain is a percept that occurs in the cortex. Please consider an alternative description such as pain behaviors, pain responses, or nociceptive responses.

4. In Figure 3, table supplement 1, two genes have superscripts that are colored red for reasons that are unclear and look peculiar.

5. In the discussion, the authors state: "We obtained ~100,000 reads per neuron which is close to the number of mRNA molecules/cell. A higher number of reads is necessary to capture rare transcripts since a single transcript can maintain ~10,000 proteins with a half-life of 1 day, which may be enough for many regulatory proteins." Please provide citations or at least some indication of how you arrived at these estimates.

6. The abbreviations associated with the PBN and its efferent targets make this paper somewhat challenging to read. Please consider adding a table of abbreviations.

7. In the results, the authors state that "This (Phox2b) line showed a unique projection to the SH, which likely originated from cells in the PBlc and PBls rather than PBmm because we did not see the same SH projection from the Tac1 cells in the PBmm" What is the SH? I could not figure this out, nor could I see evidence for this claim in the main or supplementary figures.

8. There are two undefined yet significant clusters (without any assigned color codes) – one in the center and the other at the bottom of the UMAP space (Figure 2A). The molecular identity of the two clusters should be described in the figure and main text.

9. The authors conclude that neuronal populations located in the dorsal PBN mainly innervate brain regions associated with the Central Tegmental Tract (CTT), whereas neuronal populations found in the PBle mainly innervate the brain regions associated with the Ventral Pathway (VP). However, there is significant overlap in the brain regions innervated by both PBN populations (Figure 7B). Thus, the axon projection summary diagram (Figure 7A) may be misleading. Can this be discussed and the overlap of these two pathways more clearly indicated?

*Reviewer #2 (Recommendations for the authors):*

1) Related to concern 2) in the public review: Considering the variability of the expression level of different Cre-driver lines and efficiency of AAV virus injections, quantifying the relative density of axonal projections within each population could be more meaningful and potentially better support the authors' conclusion.

*Reviewer #3 (Recommendations for the authors):*

1. In the abstract, the authors describe the PBN as being involved in pain sensation. Many people in the pain field would cringe at this description since they feel that pain is a percept that occurs in the cortex. Please consider an alternative description such as pain behaviors, pain responses, or nociceptive responses instead.

2. In Figure 3, table supplement 1, two genes have superscripts that are colored red for reasons that are unclear and look peculiar.

3. In the discussion, the authors state "We obtained ~100,000 reads per neuron which is close to the number of mRNA molecules/cell. A higher number of reads is necessary to capture rare transcripts since a single transcript can maintain ~10,000 proteins with a half-life of 1 day, which may be enough for many regulatory proteins." Please provide citations or at least some indication of how you arrived at these estimates.

4. The abbreviations associated with the PBN and its efferent targets make this paper somewhat challenging to read. Please consider adding a table of abbreviations.

5. In the results, the authors state that "This (Phox2b) line showed a unique projection to the SH, which likely originated from cells in the PBlc and PBls rather than PBmm because we did not see the same SH projection from the Tac1 cells in the PBmm" What is the SH? I could not figure this out, nor could I see evidence for this claim in the main or supplementary figures.

---

## [Author Response]

Essential revisions:1. Although both males and females were used, the authors state that "no formal comparisons were done between sex". Many of the peptides and receptors identified here exhibit sexual dimorphism. Information on potential sex differences would be valuable, and the authors should comment on this.

We added the sex of mice used in these studies to Methods and the following sentence to the Discussion (caveat section): Mice of both sexes were pooled for scRNA-Seq experiment and the Hi-Plex experiment did not have enough mice of each sex to make any formal comparison, so future studies should consider this variable.

2. Geerling and collaborators (PMC9119955) have recently published a catalog of cell populations in PB, focusing primarily, but not exclusively, on Atoh1 and Lmx1. The present authors have cited this paper more than once in this manuscript. However, it might be useful for the readers to relate the conclusions of the present manuscript to those presented by Geerling's group.

We added a sentence to Results. Karthik et al. (2022) have shown that the two major clades represented by *Atoh1* decedents and *Lmx1* descendants are largely non-overlapping populations; they have distinct axonal projections patterns, with the *Atoh1* clade following a central tegmental tract to the forebrain and the *Lmx1* clade following a ventral pathway.

3. In the abstract, the authors describe the PBN as being involved in pain sensation. Many people in the pain field would cringe at this description since they feel that pain is a percept that occurs in the cortex. Please consider an alternative description such as pain behaviors, pain responses, or nociceptive responses.

We changed text to read “nocifensive responses”

4. In Figure 3, table supplement 1, two genes have superscripts that are colored red for reasons that are unclear and look peculiar.

Color has been removed

5. In the discussion, the authors state: "We obtained ~100,000 reads per neuron which is close to the number of mRNA molecules/cell. A higher number of reads is necessary to capture rare transcripts since a single transcript can maintain ~10,000 proteins with a half-life of 1 day, which may be enough for many regulatory proteins." Please provide citations or at least some indication of how you arrived at these estimates.

A new section ‘Estimating mRNA and protein abundance per cell’ has been added to the Materials and methods to illustrate how the estimate was derived

6. The abbreviations associated with the PBN and its efferent targets make this paper somewhat challenging to read. Please consider adding a table of abbreviations.

We added a list of abbreviations

7. In the results, the authors state that "This (Phox2b) line showed a unique projection to the SH, which likely originated from cells in the PBlc and PBls rather than PBmm because we did not see the same SH projection from the Tac1 cells in the PBmm" What is the SH? I could not figure this out, nor could I see evidence for this claim in the main or supplementary figures.

We changed the sentence in the Discussion. It now reads: This line showed a projection to the septohippocampal nucleus (SH), which likely originated from cells in PBls rather than PBmm because we did not see the same SH projection from the *Tac1* injection that heavily expressed in the PBmm (compare *Phox2b* and *Tac1* whole brain expression available on Zenodo, DOI: 10.5281/zenodo.6707404).

8. There are two undefined yet significant clusters (without any assigned color codes) – one in the center and the other at the bottom of the UMAP space (Figure 2A). The molecular identity of the two clusters should be described in the figure and main text.

We added a sentence to Methods and a note of Figure legend.

“Within the neurons, two subclusters were unable to be mapped to any specific features and were excluded from the analysis (Figure 2A, gray).”

9. The authors conclude that neuronal populations located in the dorsal PBN mainly innervate brain regions associated with the Central Tegmental Tract (CTT), whereas neuronal populations found in the PBle mainly innervate the brain regions associated with the Ventral Pathway (VP). However, there is significant overlap in the brain regions innervated by both PBN populations (Figure 7B). Thus, the axon projection summary diagram (Figure 7A) may be misleading. Can this be discussed and the overlap of these two pathways more clearly indicated?

This is a good point. We added the following paragraph to Discussion.

“There is overlap of axonal projections to both pathways that probably occurs because none of the neuronal subclusters are restricted one sub-domain of the PBN. Cre-driver lines that mainly have expression in the dorsal PBN regions (*Pdyn, Tacr1, Brs3, Cbln4, Ptger3*) have axons that tend to travel through and target ventral brain regions such as the VTA, LHA, DMH, PVH, and MEPO. Lines that are categorized into the dorsal group often have fewer cells and weaker projections as a result. Some lines also have expression in PBle (*Tacr1, Ptger3*) which results in weak innervation of areas along the CTT as well. Some Cre-driver lines have strong cellular expression across most of the lateral PBN. For lines like this (*Adcyap1, Adcyap1r1, Oprm1, Crh*), there is robust expression of the AAV-driven fluorescent proteins in areas associated with the CTT such as the BNST/CEA and areas associated with the VP such as the MEPO. Overall, their projections are a combination of areas seen in the other groups.”

Reviewer #2 (Recommendations for the authors):1) Related to concern 2) in the public review: Considering the variability of the expression level of different Cre-driver lines and efficiency of AAV virus injections, quantifying the relative density of axonal projections within each population could be more meaningful and potentially better support the authors' conclusion.

Quantifying the density of projections is not worth the effort. Readers can decide for themselves by looking at primary data. We can discuss under “caveats”

Reviewer #3 (Recommendations for the authors):1. In the abstract, the authors describe the PBN as being involved in pain sensation. Many people in the pain field would cringe at this description since they feel that pain is a percept that occurs in the cortex. Please consider an alternative description such as pain behaviors, pain responses, or nociceptive responses instead.

This comment was addressed under Essential Revisions

2. In Figure 3, table supplement 1, two genes have superscripts that are colored red for reasons that are unclear and look peculiar.

This comment was addressed under Essential Revisions

3. In the discussion, the authors state "We obtained ~100,000 reads per neuron which is close to the number of mRNA molecules/cell. A higher number of reads is necessary to capture rare transcripts since a single transcript can maintain ~10,000 proteins with a half-life of 1 day, which may be enough for many regulatory proteins." Please provide citations or at least some indication of how you arrived at these estimates.

This comment was addressed under Essential Revisions

4. The abbreviations associated with the PBN and its efferent targets make this paper somewhat challenging to read. Please consider adding a table of abbreviations.5. In the results, the authors state that "This (Phox2b) line showed a unique projection to the SH, which likely originated from cells in the PBlc and PBls rather than PBmm because we did not see the same SH projection from the Tac1 cells in the PBmm" What is the SH? I could not figure this out, nor could I see evidence for this claim in the main or supplementary figures.

This comment was addressed under Essential Revisions